# LASER: LATENT SET REPRESENTATIONS FOR 3D GENERATIVE MODELING

## ABSTRACT

Neural Radiance Field (NeRF) provides unparalleled fidelity of novel view synthesis—rendering a 3D scene from an arbitrary viewpoint. NeRF requires training on a large number of views that fully cover a scene, which limits its applicability. While these issues can be addressed by learning a prior over scenes in various forms, previous approaches have been either applied to overly simple scenes or struggling to render unobserved parts. We introduce Latent Set Representations for NeRF-VAE (LASER-NV)—a generative model which achieves high modelling capacity, and which is based on a set-valued latent representation modelled by normalizing flows. Similarly to previous amortized approaches, LASER-NV learns structure from multiple scenes and is capable of fast, feed-forward inference from few views. To encourage higher rendering fidelity and consistency with observed views, LASER-NV further incorporates a geometry-informed attention mechanism over the observed views. LASER-NV further produces *diverse and plausible* completions of occluded parts of a scene while remaining consistent with observations. LASER-NV shows state-of-the-art novel-view synthesis quality when evaluated on ShapeNet and on a novel simulated City dataset, which features high uncertainty in the unobserved regions of the scene.

## 1 INTRODUCTION

Probabilistic scene modelling aims to learn stochastic models for the structure of 3D scenes, which are typically only partially observed (Eslami et al., 2018; Kosiorek et al., 2021; Burgess et al., 2019). Such models need to reason about unobserved parts of a scene in way that is consistent with the observations and the data distribution. Scenes are usually represented as latent variables, which are ideally compact and concise, yet expressive enough to describe complex data.

Such 3D scenes can be thought of as projections of light rays onto an image plane. Neural Radiance Field (NeRF, (Mildenhall et al., 2020)) exploits this structure explicitly. It represents a scene as a radiance field (a.k.a. a scene function), which maps points in space (with the corresponding camera viewing direction) to color and mass density values. We can use volumetric rendering to project these radiance fields onto any camera plane, thus obtaining an image. Unlike directly predicting images with a CNN, this rendering process respects 3D geometry principles. NeRF represents scenes as parameters of an MLP, and is trained to minimize the reconstruction error of observations from a single scene—resulting in unprecedented quality of novel view synthesis.

For generative modelling, perhaps the most valuable property of NeRF is the notion of 3D geometry embedded in the rendering process, which does not need to be learned, and which promises strong generalisation to camera poses outside the training distribution. However, since NeRF's scene representations are high dimensional MLP parameters, they are not easily amenable to generative modelling (Dupont et al., 2022). NeRF-VAE (Kosiorek et al., 2021) embeds NeRF in a generative model by conditioning the scene function on a latent vector that is inferred from a set of 'context views'. It then uses NeRF's rendering mechanism to generate outputs. While NeRF-VAE admits efficient inference of a compact latent representation, its outputs lack visual fidelity. This is not surprising, given its simple latent structure, and the inability to directly incorporate observed features. In addition, NeRF-VAE does not produce varied samples of unobserved parts of a scene.

A number of recent deterministic methods (Yu et al., 2021; Trevithick & Yang, 2021; Wang et al., 2021) uses local image features to directly condition radiance fields in 3D. This greatly improves

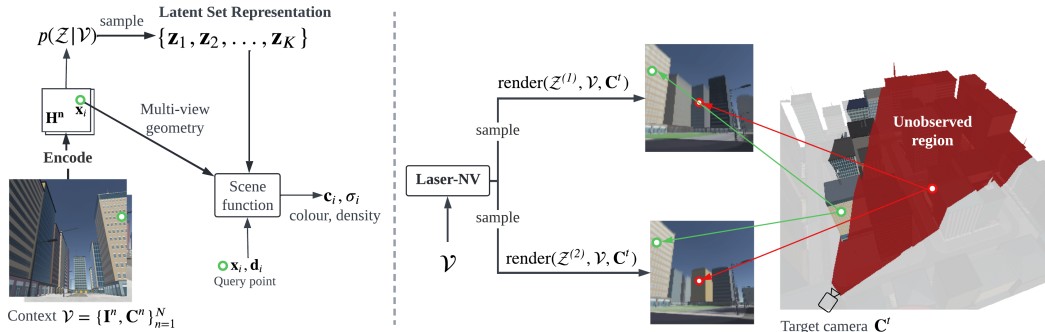

Figure 1: **Left:** LASER-NV infers a set-valued latent $\mathcal{Z}$ from the context $\mathcal{V}$ that consists of $N$ image and camera pairs $(\mathbf{I}^n, \mathbf{C}^n)$. On querying the scene function at a point $\mathbf{x}_i$ with direction $\mathbf{d}_i$, the latents are combined with local features $\mathbf{H}^n$ that are back-projected from the context views—producing color and density. **Right**: Rendering a novel viewpoint may include observed (green, an example query point on the left) and unobserved parts (red) of the scene. Conditioned on the context $\mathcal{V}$, LASER-NV allows sampling multiple scene completions that are consistent with the context views (green arrows) while providing varied explanations for the unobserved parts of the scene (red arrows). We show two such samples for the same target camera $\mathbf{C}^t$. Also see the `gif` in supp. material showing a fly-through for four prior samples conditioned on the same views.

reconstruction quality of observed parts of the scene. However, these methods are still unable to produce plausible multimodal predictions for the unobserved parts of a scene.

In this work, we address NeRF-VAE's shortcomings by proposing Latent Set Representations for NeRF-VAE (LASER-NV). To increase modelling capacity, LASER-NV uses an arbitrarily-sized set of latent variables (instead of just one vector) modelled with normalizing flows. To further enable producing samples which are consistent with observed parts, we make the generative model conditional on a set of context views (as opposed to conditioning only the approximate posterior). LASER-NV offers superior visual quality with the ability to synthesise multiple varied novel views compatible with observations. Figure 1 shows LASER-NV's key components and abilities. We include a `gif` in supp. material showing fly-throughs of additional prior samples, see Section 4.5 for details.

Our contributions are as follows:

- We introduce a novel set-valued latent representation modelled by purpose-built permutation-invariant normalizing flows conditioned on context views. We show that increasing the number of latent set elements improves modelling performance, providing a simple way to trade off computation for quality without adding new model parameters. We also verify that increasing latent dimensionality in NeRF-VAE offers no such benefits. In contrast with deterministic scene models in the literature, our probabilistic treatment over the latent set allows covering multiple models when predicting novel views.

- We develop a novel attention mechanism to condition the scene function on the set-valued latent as well as additional local features computed from context views. We show that including local features further improves visual quality.

- We evaluate LASER-NV on three datasets: a category-agnostic ShapeNet dataset, Multi-ShapeNet, and on a novel "City" dataset that contains a large simulated urban area and poses significant challenges as a benchmark for novel view synthesis due to high uncertainty in the unobserved parts of the scene. Our model overcomes some of the main limitations of NeRF-VAE and also outperforms deterministic NeRF models on novel view synthesis in the face of uncertainty.

## 2 BACKGROUND: NERF & NERF-VAE

Neural Radiance Field (NeRF, (Mildenhall et al., 2020)) represents a 3D scene as a *scene function* $F(\boldsymbol{x}, \boldsymbol{d})$ of a point coordinate $\boldsymbol{x} \in \mathbb{R}^3$ and direction $\boldsymbol{d} \in \mathbb{R}^3$, and outputs the colour $\boldsymbol{c} \in \mathbb{R}^3$ and volume density $\sigma \geq 0$. This function is parameterized using a neural network (typically a fully connected MLP). To obtain a pixel colour, the scene function is evaluated at many points along the corresponding ray in the volumetric rendering process.

This volumetric rendering integral is in practice approximated using numerical integration; see (Mildenhall et al., 2020; Blinn, 1982) for details. We use $\hat{\boldsymbol{I}} = \texttt{render}(F(\cdot), \boldsymbol{C})$ to denote the image rendering process that outputs the image $\hat{\boldsymbol{I}} \in \mathbb{R}^{H \times W \times 3}$ for the rays of a camera $\boldsymbol{C}$ and a scene function $F$. The camera $\boldsymbol{C} = (\boldsymbol{K}, \boldsymbol{R}, \boldsymbol{t})$ is specified by its intrinsic parameters $\boldsymbol{K} \in \mathbb{R}^{3 \times 3}$ and extrinsic parameters given by a rotation $\boldsymbol{R} \in \mathbb{R}^{3 \times 3}$ and translation $\boldsymbol{t} \in \mathbb{R}^3$. In its original formulation, the parameters of NeRF are optimized by minimizing an error function (typically the mean-squared error) between rendered images and the ground-truth images across many views of a single scene.

While NeRF learns high-fidelity scene representations when many views are available, it does not learn a prior over scenes, does not provide compact scene representations, and does not admit efficient inference (i.e. fast estimation of the scene parameters from a few input views). To address these issues, NeRF-VAE (Kosiorek et al., 2021) embeds NeRF as a decoder in a variational auto-encoder (VAE, (Kingma & Welling, 2014; Rezende et al., 2014)). Each scene is represented by a latent vector $\boldsymbol{z}$ (instead of the set of parameters of an MLP as in NeRF) which can be either sampled from a Gaussian prior $p(\boldsymbol{z})$ or from the approximate posterior $q(\boldsymbol{z} \mid \boldsymbol{C})$ inferred from a set of context views and associated cameras $\boldsymbol{C}$. The latent $\boldsymbol{z}$ is used to condition an MLP that parameterizes the scene function, $F_\theta(\cdot, \boldsymbol{z}) : (\boldsymbol{x}_i, \boldsymbol{d}_i) \mapsto \boldsymbol{c}_i, \boldsymbol{\sigma}_i$. Here, parameters $\theta$ are shared between scenes, while $\boldsymbol{z}$ is scene specific. NeRF-VAE uses a Gaussian likelihood $p(\boldsymbol{I} \mid \boldsymbol{z}, \boldsymbol{C})$ with the image rendered from the sampled scene function as mean, and a fixed standard deviation.

## 3 LATENT SET REPRESENTATIONS FOR NERF-VAE

LASER-NV is a conditional generative model, conditioned on a set of input views $\mathcal{V} := \{\boldsymbol{I}^n, \boldsymbol{C}^n\}_{n=1}^N$. Given these views the model infers a latent representation that can be used for novel view synthesis.

In order to represent large-scale and complex scenes we use **La**tent **Se**t **R**epresentations (LASERs), a set of latents of the form $\mathcal{Z} := \{\boldsymbol{z}_k\}_{k=1}^K$, where $K$ is a hyperparameter. In the NeRF-VAE, the scene function MLP is directly conditioned on the inferred latent. In contrast, LASER-NV's scene function is conditioned on the latent set $\mathcal{Z}$ and features $\mathcal{H} := \{\boldsymbol{H}^n\}_{n=1}^N$ extracted from $N$ input views, and integrates information from both. This results in the form $F_\theta(\cdot, \mathcal{Z}, \mathcal{H}) : (\boldsymbol{x}_i, \boldsymbol{d_i}) \mapsto \boldsymbol{c}_i, \boldsymbol{\sigma}_i$, which we detail further in Section 3.2.

We obtain the input features $\mathcal{H}$ by separately encoding the input views using a convolutional neural net (CNN) that maintains spatial structure. These feature maps are subsequently used in the conditional prior, the posterior, and the resulting scene function, all of which we describe next. Further architectural details are provided in Appendix A.

### 3.1 CONDITIONAL PRIOR AND POSTERIOR

Given a set of context views $\mathcal{V} := \{\boldsymbol{I}^n, \boldsymbol{C}^n\}_{n=1}^N$ consisting of images $\boldsymbol{I}^n$ and corresponding cameras $\boldsymbol{C}^n$, LASER-NV defines a conditional prior $p(\mathcal{Z}|\mathcal{V})$ over a latent set of size $K$, parameterized by a permutation-invariant normalizing flow— a distribution model that allows for both fast and exact sampling and efficient density evaluation (Rezende & Mohamed, 2015; Papamakarios et al., 2021).

Flows are defined by an invertible mapping $f$ of random variables, starting from a base distribution $p_0(\mathcal{Z}^{(0)})$. When $f$ is composed of multiple invertible mappings $f_1, \ldots, f_m$, the resulting density is given by

$$p(\mathcal{Z} \mid \mathcal{V}) = p_0(f^{-1}(\mathcal{Z}, \mathcal{V})) \prod_{m=1}^M \left| \det \frac{\delta f_m(\mathcal{Z}^{(m-1)}, \mathcal{V})}{\delta \mathcal{Z}^{(m-1)}} \right|^{-1}. \tag{1}$$

Our flow design closely follows that of Wirnsberger et al. (2020; 2021), who parameterize the transformations in each layer using transformers, preserving permutation invariance. Specifically, we propose a novel design to condition the flow distribution on the features $\mathcal{H}$ extracted from the context views. To achieve that, we use split coupling layers (Dinh et al., 2017), where for each layer, we first apply self-attention to the set of latents, followed by cross-attention to attend the context features. The output of the second transformer defines the parameters for an invertible affine transformation of the random variables. Implementation details of our normalizing flow is in Appendix A.1.

Since LASER-NV is trained as a VAE, we define an *approximate posterior* $q(\mathcal{Z}|\mathcal{U} \cup \mathcal{V})$, where $\mathcal{U} = \{\boldsymbol{I}^t, \boldsymbol{C}^t\}_{t=1}^M$ are target views that the model is trained to reconstruct, denoted as posterior context. The posterior and conditional prior are both distributions over the latent space, where the posterior is conditioned on the union of prior and posterior context. We model the posterior distribution using the same architecture as the conditional prior, but with a separate set of learned parameters.

## 3.2 Decoder & Scene Function

When evaluating the scene function, i. e. computing colour and density for a point $\mathbf{x}_i, \mathbf{d}_i$, LASER-NV has access to both the latent set $\mathcal{Z}$ as well as the features $\mathcal{H}$ extracted from the context views $\mathcal{V}$. We now describe how to integrate those two, which consists of querying the latent set for each point, and using 3D projection to extract relevant features.

**Querying the Latent Set**   While the latent set representation $\mathcal{Z}$ contains information regarding the entire scene, when querying for a specific point $\boldsymbol{x}_i$, we would like to extract only relevant information from $\mathcal{Z}$. We propose using a transformer cross-attention model to compute a single feature vector from $\mathcal{Z}$. In the first layer, queries are computed from the positional encoding of $\boldsymbol{x}_i$, and keys and values are computed from the elements of $\mathcal{Z}$. In subsequent layers, queries are computed from previous layers' output. We denote the output of the latent querying as latent features $\tilde{\boldsymbol{z}}_i$, each corresponding to a query point $\boldsymbol{x}_i$.

**Local Features**   When synthesizing a novel view given a few input views, a number of existing NeRF-based methods uses 3D projection to determine which features from the context views are useful for explaining a particular 3D point (Yu et al., 2021; Trevithick & Yang, 2021; Wang et al., 2021). LASER-NV incorporates this idea, and in particular follows the design of pixelNeRF (Yu et al., 2021) as it is shown to achieve strong results in a wide range of domains.

We use the encoded input views $\{\boldsymbol{H}^n\}_{n=1}^N$ described above, along with their corresponding camera matrices $\{\boldsymbol{C}^n\}_{n=1}^N$. Given a query 3D point $\boldsymbol{x}_i$ and direction $\boldsymbol{d}_i$, we obtain the point $\boldsymbol{p}_i^n \in \mathbb{R}^2$ in the image space of the $n^{\text{th}}$ context view using the known intrinsic parameters. The corresponding local features are then bilinearly interpolated $\boldsymbol{h}_i^n = \boldsymbol{H}^n[\boldsymbol{p}_i^n]$. We refer to $\boldsymbol{h}_i$ as *local features* since they are spatially localized based on the projected coordinates, as opposed to being a global aggregation of all context features. We finally obtain $\hat{\boldsymbol{h}}_i^n$ for each context view by separately processing the features $\boldsymbol{h}_i^n$ using a residual MLP that has as an additional conditioning on the image-space projected points and directions.

**Integrating Latents and Local Features**   In contrast to pixelNeRF, which does not have latents, LASER-NV integrates the both local and the latent features. It does so with a transformer:

$$\boldsymbol{f}_i = M_\theta(\tilde{\boldsymbol{z}}_i; \bigcup_{n=1}^N \{\boldsymbol{h}_i^n : \text{is\_visible}(\mathbf{x}_i, \mathbf{C}^n)\} \cup \{\tilde{\boldsymbol{z}}_i\}) \tag{2}$$

The attention queries are computed from latent features $\tilde{\boldsymbol{z}}_i$ (see above), and attend to the combined set of local features and to $\tilde{\boldsymbol{z}}_i$. Note that, for a point $\mathbf{x}_i$, we use only those context views from which this point is visible. Using a transformer to process encoded features from multiple views with epipolar constraints has been proposed in prior work on Multi-view Stereo, see e.g. (He et al., 2020; Xi et al., 2022; Wang et al., 2022; Ding et al., 2022).

The final step in evaluating the scene function is to use $\boldsymbol{f}_i$ to condition a residual MLP similar to NeRF-VAE resulting in LASER-NV's scene function $F_\theta(\cdot, \mathcal{Z}, \mathcal{H}) : (\mathbf{x}_i, \mathbf{d}_i) \mapsto \boldsymbol{c}_i, \sigma_i$ which is used to volume-render image outputs.

## 3.3 Loss

Similarly to NeRF-VAE, our objective function is given by the evidence lower-bound:

$$\mathcal{L}(\mathcal{U}, \mathcal{V}) = \mathbb{E}_{\mathcal{Z} \sim q} \left[ \sum_{t=1}^{M} \log p\big(\boldsymbol{I}^t \mid \boldsymbol{C}^t, \mathcal{Z}, \mathcal{V}\big) \right] - \mathrm{KL}(q(\mathcal{Z} \mid \mathcal{U} \cup \mathcal{V}) \mid\mid p(\mathcal{Z} \mid \mathcal{V})) \qquad (3)$$

where $\boldsymbol{I}^t$ and $\boldsymbol{C}^t$ are the images and cameras from the posterior context views $\mathcal{U}$. We use a Gaussian likelihood term for $p(\boldsymbol{I}^t \mid \boldsymbol{C}^t, \mathcal{Z}, \mathcal{V})$ with a fixed variance.

## 4 Experiments

### 4.1 Datasets

We first evaluate LASER-NV on a novel dataset based on a set of synthetic, procedurally generated, environments consisting of urban outdoor scenes. This dataset contains posed images of a virtual city with objects of multiple scales, such as house blocks, roads, buildings, street lights, etc (but no transient objects such as cars or people). As such, it is an extremely challenging scene representation benchmark, especially when representations are inferred from few views only—resulting in partial observability and high uncertainty. In this situation, when evaluated on unobserved parts of a scene, we want a model to produce predictions that are varied, self-consistent, and aligned with the data distribution. Each scene consists of a neighborhood block, and a randomly chosen designated point of interest (road intersection or park) is selected within. Images are then rendered by placing the camera at random nearby positions and viewpoints from the point of interest. The dataset has 100K training scenes, and 3200 test scenes. We provide further details and dataset samples in Appendix B.

To test LASER-NV's ability to produce near-deterministic scene completions in a low-uncertainty setting, we compare it against a number of previous methods on the ShapeNet NMR dataset (Kato et al., 2018), which consists of several views of single objects from 13 different ShapeNet categories. Finally, we evaluate on the more challenging MultiShapeNet-Hard (MSN-Hard) (Sajjadi et al., 2021), which tests the models capabilities in the presence of a large number of cluttered objects of varying size with detailed textures and realistic backgrounds.

### 4.2 Models & Evaluation

On the City dataset, we compare LASER-NV with Multi-view Conditional NeRF (MVC-NeRF) and NeRF-VAE; see Figure 7 in Appendix A for model diagrams. MVC-NeRF (Figure 7c) is a variant of LASER-NV (Figure 7a) which does not use LASER-NV's latent set representations but instead only relies on deterministically rendering new views using local features. It therefore closely resembles the design of pixelNeRF but uses the same architecture for the image encoder and scene function $F_\theta$ as LASER-NV to ensure they are comparable. On ShapeNet, we further show quantitative results of a number of non-NeRF novel view synthesis methods: Scene Representation Networks (SRN, Sitzmann et al. (2019)), Differentiable Volumetric Rendering (DVR, Niemeyer et al. (2020)), and Scene Representation Transformers (SRT, Sajjadi et al. (2021)). During training, LASER-NV uses a random subset of 4 of the 23 target views as posterior context views[1]. MVC-NeRF and NeRF-VAE are trained to directly predict the 23 views given a single context view. At test time, LASER-NV samples the latents from its conditional prior whereas NeRF-VAE uses its posterior (given that its prior is unconditional). Importantly, all models in this evaluation use one input view and predict 23 target views.

At evaluation time, we feed each model with a set of context views, and evaluate *reconstructions* of those same views, and *predictions* of novel views of the same scene. We evaluate reconstruction ability via likelihood estimates, and standard image similarity metrics PSNR, and SSIM (Wang et al., 2004). For novel view predictions we report the Fréchet Inception Distance (FID, Heusel et al. (2017)) between the marginal distribution of samples and the evaluation dataset[2]. This metric

---

[1] While using the 23 views as posterior context would be the standard set up (Bayer et al., 2021), we find that using 4 randomly-sampled views is much cheaper computationally and yet sufficient for training.

[2] Image similarity metrics such as PSNR and SSIM are not useful when there exists a large multimodal space of possible predictions as is the case in the City.

highlights the incapability of deterministic models to generate varied and plausible samples in face of uncertainty. Note that since NeRF-VAE is unconditional, we use the learned approximate posterior as the conditional distribution. In the City, we thus train a separate model (one to use for reconstruction metrics that has 8 posterior context views and another one with 2 views for predictions). For ShapeNet results all models use one context view.

On MSN-Hard, we train LASER-NV with 5 prior context views and 3 target views. We report the test PSNR of 1 novel view given 5 input views, and compare with the results reported in Sajjadi et al. (2022). The methods of comparison include two amortized non-NeRF models based on light fields, Scene Representation Transformers (SRT) and Object SRT (OSRT Sajjadi et al. (2022)); and two amortized NeRF methods, pixelNeRF and Volumetric OSRT (VOSRT).

### 4.3 ABLATIONS

We further ablate LASER-NV to isolate contributions of individual model components. To quantify the effects of conditioning the prior we compare LASER-NV to a conditional version of NeRF-VAE. Note that the original NeRF-VAE used an unconditional prior (Kosiorek et al., 2021). We also compare LASER-NV to a version that does not condition the scene function with local context features, cf. Equation (2) and Figure 7b in Appendix A, denoted by LASER-NV (NO GEOM.). LASER-NV, LASER-NV (NO GEOM.), NeRF-VAE variants, and MVC-NeRF share architectures for the image encoder and the scene function $F_\theta$.

### 4.4 TRAINING AND IMPLEMENTATION DETAILS

We use the hierarchical sampling technique of (Mildenhall et al., 2020) when rendering a pixel colour, which involves learning two separate sets of scene function parameters as well as optimizing a corresponding colour likelihood term for each step. For experiments with the City, we leverage ground-truth depth maps using the method proposed by Stelzner et al. (2021) in order to train all the models with fewer scene function evaluations per ray. Not using ground-truth depth significantly increases training time while only marginally lowering reconstruction PSNR, see Appendix D for a discussion and Fig. 16 in Appendix B for visualisations. Importantly, when evaluating our models we revert to volumetric rendering. All generative models are trained by optimizing the evidence lower bound; where we anneal the KL term in Eq. (3). All experiments use the Adam (Kingma & Ba, 2014) optimizer. We provide full details of model architectures in Appendix A and training procedures in Appendix E.

### 4.5 RESULTS

Results on the City dataset are shown in Table 1. In terms of reconstructing observed input views, both LASER-NV and MVC-NeRF, due to the use of explicit geometrical knowledge, result in excellent reconstructions. NeRF-VAE, however, needs to reconstruct input views purely from its latent representation, resulting in low fidelity reconstructions. For prediction of unseen views, the output distribution of LASER-NV is closer to the actual evaluation data (lower FID), in contrast to MVC-NeRF which outputs a single prediction, and NeRF-VAE which produces low quality outputs that do not vary much. We show representative example reconstructions and predictions in Fig. 2. We also include a `gif`[3] in supp. material showing fly-throughs of model samples.

Results on ShapeNet are shown in Table 2. Both MVC-NeRF and LASER-NV outperform existing published methods when predicting novel views of these single-object scenes. We further observe low quality predictions of NeRF-VAE in example predictions in Fig. 11. Finally, Fig. 12 shows a hand-selected example where the input view has ambiguity and how LASER-NV can sample plausible variations. As in the City experiment, LASER-NV produces plausibly varied predictions of the object in contrast to the other methods.

Results on MSN-Hard are shown in Table 3. LASER-NV clearly outpeforms the NeRF-based models, and improves over OSRT and the original SRT. Only a modified SRT shows higher PSNR (25.93) over LASER-NV (24.45). Note that SRT does not learn to estimate densities nor depth maps and

---

[3]To create this visualisation we trained the model on a modified City dataset with birds-eye views.

|  | Reconstruction | | | Prediction |
| --- | --- | --- | --- | --- |
|  | Log-Likelihood ↑ | PSNR↑ | SSIM↑ | FID↓ |
| MVC-NeRF | - | 28.63 | 0.910 | 91.92 |
| Cond. NeRF-VAE | 3.51 | 23.94 | 0.709 | 50.98 |
| NeRF-VAE | 3.50 | 23.92 | 0.708 | 52.12 |
| LASER-NV (NO GEOM.) | **3.82** | 27.13 | 0.870 | 24.19 |
| **LASER-NV** | **3.83** | **29.61** | **0.923** | **22.54** |

Table 1: Results in City. The color likelihood is computed via importance sampling (10 samples); note that MVC-NeRF is deterministic and thus we do not compare it on this metric. **Reconstruction**: We report performance of reconstruction of the 2 context views. **Prediction**: We evaluate the models on previously unobserved views using the FID score. MVC-NeRF provides a single deterministic prediction whereas for the other models we average over 10 independent samples.

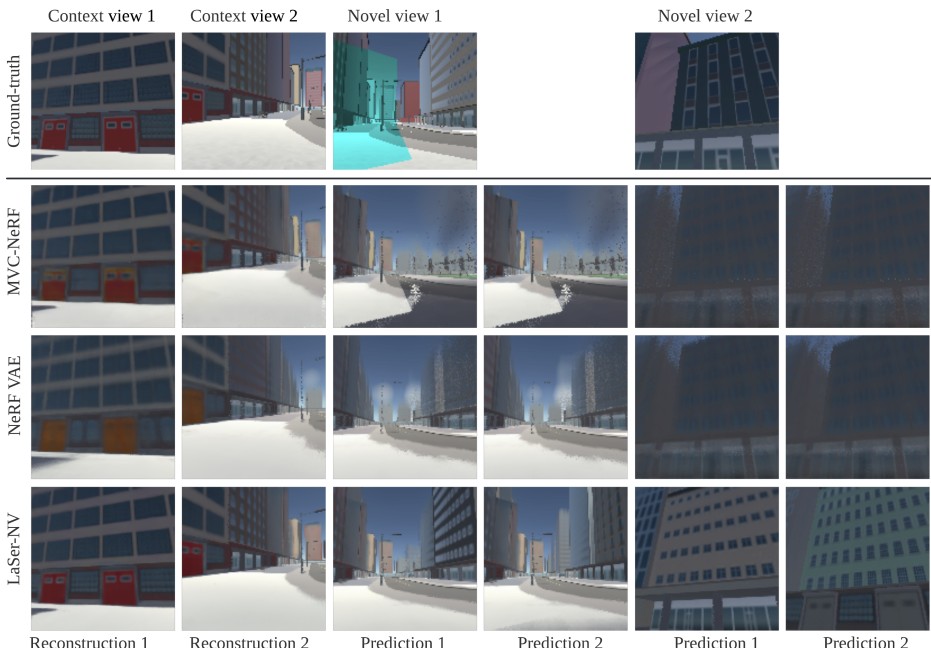

Figure 2: Representative reconstructions of observed views and predictions of partially unobserved novel views. The models are conditioned on two context views (top left). The first two columns show reconstructions of those views. Note how NeRF-VAE struggles to reconstruct both views accurately, while the other models do well. We show predictions of two novel views (top right), where the first view is partially observed in the context (highlighted in cyan color) and the second view is fully unobserved. LASER-NV's predictions are consistent with the observed part of the context, diverse, and plausibly so. NeRF-VAE's predictions lack quality and diversity. MVC-NeRF's only produces a single prediction, which is not very plausible.

is deterministic. Fig. 3 shows a prediction example using LASER-NV, and further examples are in Appendix C which demonstrate high fidelity colours and densities.

### 4.6 TRAIN- AND TEST-TIME SCALING & DATA EFFICIENCY

Latent set representations allow trading computation for increased model capacity. We train LASER-NV on the City dataset with increasing latent set sizes $K$, and compare it to a conditional NeRF-VAE also trained with increasing latent dimensions[4]. Figure 4 shows reconstruction log-likelihoods as a

---

[4]Note that this increases the number of NeRF-VAE's parameters.

|          | DVR*  | SRN*  | SRT*  | pixelNeRF* | NeRF-VAE | MVC-NeRF | LASER-NV |
|----------|-------|-------|-------|------------|----------|----------|----------|
| PSNR ↑   | 22.70 | 23.28 | 27.87 | 26.80      | 25.36    | **28.03**| 27.92    |
| SSIM ↑   | 0.860 | 0.849 | 0.912 | 0.910      | 0.875    | **0.925**| **0.923**|

Table 2: Category-agnostic ShapeNet evaluation by reconstructing 23 views from one input view for all models. We further report numbers from the respective papers for DVR* (Niemeyer et al., 2020), SRN* (Sitzmann et al., 2019), SRT* (Sajjadi et al., 2021), and pixelNeRF* (Yu et al., 2021).

|        |          | NeRF  |          |      | Light field |       |
|--------|----------|-------|----------|------|------|-------|
|        | pixelNeRF | VOSRT | LASER-NV | OSRT | SRT  | SRT++ |
| PSNR ↑ | 21.97    | 21.38 | **24.45**| 23.33 | 23.54 | **25.93** |

Table 3: Novel view PSNR of one view from 5 input views on MSN-Hard. PixelNeRF results are from Sajjadi et al. (2021) and the remaining from Sajjadi et al. (2022). SRT++ is an improved SRT (Sajjadi et al., 2022).

| Inputs | GT | Prediction | Depth |

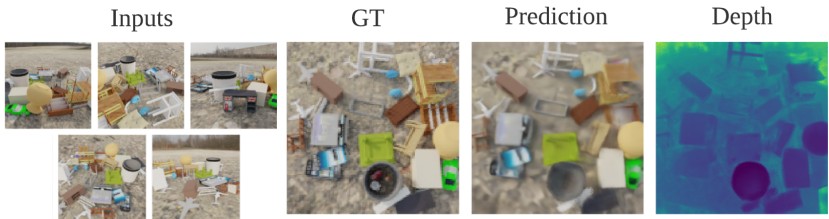

Figure 3: Novel view syntesis with LASER-NV on MSN-Hard.

function of latent capacity. While LASER-NV consistently improves its reconstructions with larger latent sets, cond. NeRF-VAE quickly saturates at 256 latent dimensions, at worse reconstructions compared to LASER-NV. Further, in Appendix D, we provide an analysis of test-time scaling—we show that increasing the latent set size at test time does not result in clear change of performance. Finally, in Figure 14 we show that LASER-NV performs better at test time than NeRF-VAE and MVC-NeRF for different training dataset sizes.

## 5 RELATED WORK

**Neural scene representations**   Inferring neural scene representations using a deep generative model of scenes is an active research area, and we mention some of the most closely related approaches below. NeRF-VAE learns a generative model of 3D scenes which is only shown to work in relatively simple scenes. When applied to more complicated data (see Fig. 2), it produces blurry reconstructions and novel views. We posit that this is caused by insufficient capacity of its latent representation $z$. Generative Query Network (Eslami et al., 2018) similarly learns a distribution over scenes, but its decoder is a CNN. Subsequent work (Rosenbaum et al., 2018) improves over its representational capacity by using an attentive mechanism when conditioning the decoder from context in an analogous way to how LASER-NV improves over NeRF-VAE. By implementing a NeRF in our decoder we gain the benefits of more robust generalization capabilities to novel viewpoints compared to the convolutional counterparts (Kosiorek et al., 2021). Simone (Kabra et al., 2021) is a convolutional generative model that performs high quality novel view synthesis and while also performing unsupervised segmentation of input videos. The authors in (Dupont et al., 2022) propose directly modelling the underlying radiance fields as functions, which they refer to as 'functa'. Both our method and theirs learn a prior over scene functions with a model that has a large representational capacity. Bautista et al. (2022) propose a two-stage method to learn a generative model of 3D scenes. As LASER-NV, their model can conditionally sample radiance fields. However, they focus on scenes with dense views, whereas LASER-NV works with sparse views.

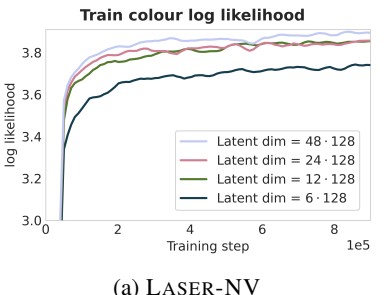
(a) LASER-NV

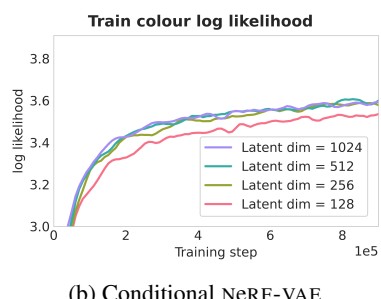
(b) Conditional NeRF-VAE

Figure 4: Training reconstruction performance in City of LASER-NV with an increasing latent set size (each with 128 dimensions) vs a cond. NeRF-VAE with increasing latent dimensionality.

**Deterministic novel view synthesis**   NeRF relies on optimizing the parameters of an MLP scene function, separately for every scene. A number of methods in the literature propose amortizing the mapping of input views to the scene function using multi-view geometry, see e.g. IBRNet (Wang et al., 2021), General Radiance Fields (GRF) (Trevithick & Yang, 2021) and pixelNeRF (Yu et al., 2021). These methods extract features of the context views by using projective geometry to select which features to attend to, and are shown to generalize well and carry out high-fidelity predictions. PixelNeRF is designed to work well in a solely view-centered coordinate system. MVSNerf (Chen et al., 2021) lifts context features to a 3D representation with strong novel-view synthesis performance. Scene Representation Transformers (SRT) (Sajjadi et al., 2021) use global features obtained from context views using a generalization of the Vision Transformer (Dosovitskiy et al., 2020), which bears similarities to LASER-NV. SRT bypasses the use of 3D geometry entirely, yet shows remarkable performance on real world data.

**NeRF for large-scale scenes**   A number of NeRF methods have been developed to scale up to city-wide scenes. They work by optimizing multiple separate scene functions for separate blocks of a scene, which are then dynamically used when rendering (Tancik et al., 2022; Turki et al., 2021). Alternatively they design a scene function that represents the scene at multiple levels of scale (Martel et al., 2021; Xiangli et al., 2021). Similarly, Mip-NeRF (Barron et al., 2021) allows prefiltering the scene function inputs to better capture coarse and fine details. These methods are orthogonal to ours since they do not learn a conditional NeRF model sparse views nor address the uncertainty in predictions.

## 6   CONCLUSION

We propose LASER-NV, a conditional generative model of neural radiance fields capable of efficient inference of large and complex scenes under partial observability conditions. While recent advances in neural scene representation and neural rendering lead to unprecedented novel view synthesis capabilities, producing sharp and multi-modal completions of unobserved parts of the scene remains a challenging problem. We experimentally show that LASER-NV can model scenes of different scale and uncertainty structure, and isolate the usefulness of each contribution through ablation studies. While NeRF serves as a strong inductive bias for learning 3D structure from images, LASER-NV also inherits some of its drawbacks. Volumetric rendering with neural radiance fields is computationally costly and can limit the model from real-time rendering. It remains to be seen whether fast NeRF implementations (Müller et al., 2022) can be of help. Furthermore, accurate GT camera information is required for learning and novel view synthesis; (Moreau et al., 2022) recently made progress on localization methods using NeRF. While learning a generative scene model of real scenes remains an open problem, we believe this work is an important step in that direction. Finally, an interesting direction for future research is incorporating object-centric structure and dynamics into LASER-NV, which might be useful for for downstream reasoning tasks (Sajjadi et al., 2022; Ding et al., 2021).

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

# A  MODEL ARCHITECTURES

## A.1  NORMALIZING FLOWS FOR LASER

We wish to model our latent set representation $\mathcal{Z} = \{z_1, \ldots, z_K\}$ using a flexible distribution that is efficient both to evaluate as well as sample from. Furthermore, the distribution should ideally be invariant to re-ordering of the elements of the set. Normalizing flows (Rezende & Mohamed, 2015) allow specifying a distribution with precisely these properties. Formally, it is given by:

$$\mathcal{Z}^{(0)} \sim p_0(\mathcal{Z}), \ \mathcal{Z}^{(m)} = f_m\Big(\mathcal{Z}^{(m-1)}, \mathcal{V}\Big), \ m = 1, \ldots, M, \ \text{with } \mathcal{Z} \coloneqq \mathcal{Z}^{(M)}, \tag{4}$$

where $p_0(\cdot)$ is a base distribution, $f_m(\cdot)$ are invertible transformations applied to an intermediate random variable, and $\mathcal{Z}$ is the final LASER after $M$ flow layers. $\mathcal{V}$ is an additional set of context vectors that condition the flow transformation. The resulting probability density of $\mathcal{Z}$ is

$$p(\mathcal{Z} \mid \mathcal{V}) = p_0\Big(\mathcal{Z}^{(0)}\Big) \prod_{m=1}^{M} \left| \det \frac{\delta f_m\big(\mathcal{Z}^{(m-1)}, \mathcal{V}\big)}{\delta \mathcal{Z}^{(m-1)}} \right|^{-1}. \tag{5}$$

The need to compute the Jacobian of intermediate flow transformations puts constraints on the design of those transformations, as they need to be computationally efficient. We use a split-coupling flow similar to RealNVP (Dinh et al., 2017), and we follow a similar architecture as the ones used in Wirnsberger et al. (2020; 2021) in adapting it to sets. The method Flow Scans (Bender et al., 2020) first introduced the general idea of adapting split-coupling to work on sets, and Wirnsberger et al. (2020; 2021) use a transformer for increased flexibility. A split-coupling transformation partitions the set $\mathcal{Z}$ into two sets $\mathcal{Z}_1, \mathcal{Z}_2$ by splitting each element of the original set across channels (we split in half). At a high level, the transformation of one part is computed as a function of the other part. Finally, the original set is recovered by elementwise concatenation of both transformed parts. More formally, let $g$ be an invertible function, a split-coupling transform $f_m^{sc}$ that maps $\mathcal{Z}^{(m-1)} \to \mathcal{Z}^{(m)}$ is given by three operations:

$$\begin{aligned}
\mathcal{Z}_1^{(m)} &= g\Big(\mathcal{Z}_1^{(m-1)}, t_m\Big(\mathcal{Z}_2^{(m-1)}, \mathcal{V}\Big)\Big), \\
\mathcal{Z}_2^{(m)} &= g\Big(\mathcal{Z}_2^{(m-1)}, t_m\Big(\mathcal{Z}_1^{(m)}, \mathcal{V}\Big)\Big), \\
\mathcal{Z}^{(m)} &= \{\text{concat}(z_{k,1}^{(m)}, z_{k,2}^{(m)})\}_{k=1}^{K},
\end{aligned} \tag{6}$$

where $z_{k,1}$ indicates dimensions of split 1 of element $k$, and $\mathcal{V}$ is conditioning context. Note how the parameters of the transformation $g$ of one partition are derived from the other partition. To do so in a permutation-equivariant manner, the functions $t^m$ are a composition of two transformers: a self-attention module is applied followed by a cross-attention conditioned on context features, defined in Appendix A.2. More specifically, when transforming split 1 as a function of split 2, we have:

$$\psi_2^{(m-1)} = \text{CrossAttn}\Big(\text{SelfAttn}\Big(\mathcal{Z}_2^{(m-1)}\Big), \mathcal{V}\Big). \tag{7}$$

We finally apply the function $g$, in our case the affine function (Dinh et al., 2017), from the parameters obtained for every element in $\psi_2^{m-1}$:

$$z_{k,1}^{(m)} = z_{k,1}^{(m-1)} \exp(\boldsymbol{W}_{scale}^{m} \psi_{k,2}^{(m-1)}) + \boldsymbol{W}_{scale}^{m} \psi_{k,2}^{(m-1)} \text{ for } k = 1, \ldots K, \tag{8}$$

where $\boldsymbol{W}_{scale}^{m}$ and $\boldsymbol{W}_{bias}^{m}$ are linear weights. When composing our complete flow with $M$ layers of split-coupling flows, we find that including an additional linear invertible transformation after each $f_m$ to be helpful. To do so, we use the *Linear Permutation Equivariant* ($f_m^{lpe}$), proposed in the Flow Scans work, the purpose of which is to allow capturing interdependencies in the dimensions of the set elements. The only difference is that we parameterize each sub-block of the linear transformation using a full unconstrained matrix (instead of a diagonal matrix).

In summary, our flow chains together the sequence of mappings $f_1^{sc}, f_1^{lpe}, \ldots, f_M^{sc}, f_M^{lpe}$. Thus, sampling from this flow involves sampling from the base distribution followed by applying the chain of functions.

Our normalizing flow differs in a number of ways compared with Wirnsberger et al. (2020; 2021). First, our flow layers are additionally parameterized by a cross-attention module in order to condition the distribution on the context set; second, we don't use rational-quadratic splines, instead we found the simpler affine coupling layer to work well. Notice that the size of the resulting set depends only on the hyper-parameter $K$, that is the number of elements sampled from the base distribution. It does not depend on the context size, i.e. features extracted from input images.

In our experiments, we use a diagonal Gaussian base distribution, and set the number of flow layers to be $M = 8$ for the prior flow, and $M = 4$ for the posterior flows. Our self-attention and cross-attention are composed of $L = 1$ layers, and 256 hidden units. Furthermore, we invert the chain of mappings (which is possible due to invertibility) for the prior flow as it we found it to be more numerically stable during training.

## A.2 ATTENTION MODULES

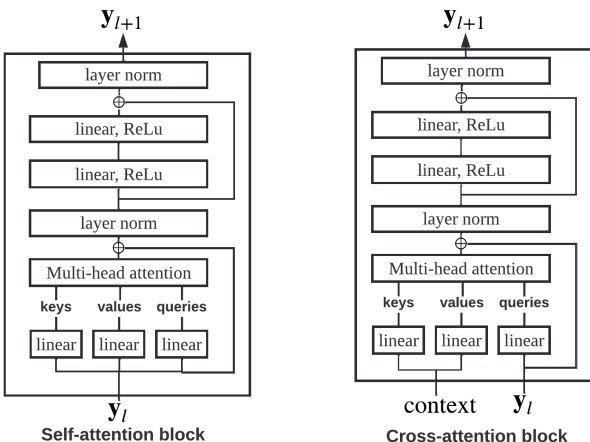

Figure 5: Multi-head self-attention and cross-attention blocks. When combining multiple blocks, the outputs $\boldsymbol{y}_l$ for layer $l$ become the inputs for layer $l + 1$.

We use transformer attention Vaswani et al. (2017) in various components of LASER-NV. We distinguish two types of attention modules: self-attention, where keys, queries and values are computed from the same input, and cross-attention, where some context is used to compute keys and values, and queries are computed from a different input). An attention module can be composed of multiple attention blocks, where each attention block is shown in Fig. 5. We use the standard multi-head dot-product attention with softmax weights. We define SelfAttn$(\boldsymbol{y}, L)$ to be a self-attention module with input $\boldsymbol{y}$ and $L$ blocks[5] and, similarly, CrossAttn$(\boldsymbol{y}, \boldsymbol{c}, L)$ to be a cross-attention module with context $\boldsymbol{c}$. Note how SelfAttn and CrossAttn do not include positional encoding that are often used in transformers.

## A.3 IMAGE ENCODER

To encode the context images, we use a convolutional neural network based on the ResNet architecture from Vahdat & Kautz (2020) (also described in Kosiorek et al. (2021)) to independently encode context elements into $h \times w$ feature maps. The residual blocks that compose the encoder are shown in Fig. 6. For all models and datasets we use $L = 4$. Each context view consists of an image in RGB space with concatenated information of the camera position and direction.

For $N$ context views, our last layer outputs $Nhw$ vectors that form the conditioning set used to parameterize our distributions over $\mathcal{Z}$. Furthermore, the feature maps at different layers are also combined into a feature stack to use as the context local features as described in Appendix A.4.

---

[5]We often drop the $L$ input argument for notational clarity.

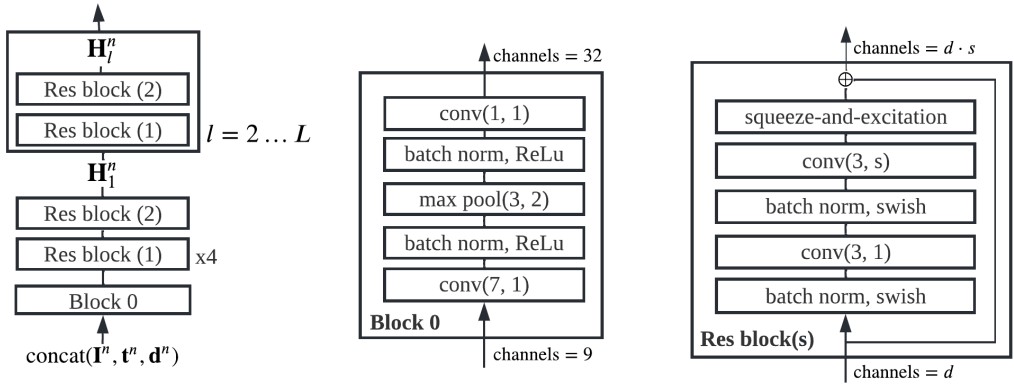

Figure 6: ResNet architecture based on the encoder of NouveauVAE Vahdat & Kautz (2020).

## A.4 SCENE FUNCTION ARCHITECTURES

Fig. 7 provides a high-level overview of the information flows going into the scene function of each model, and we describe implementation details of their components below.

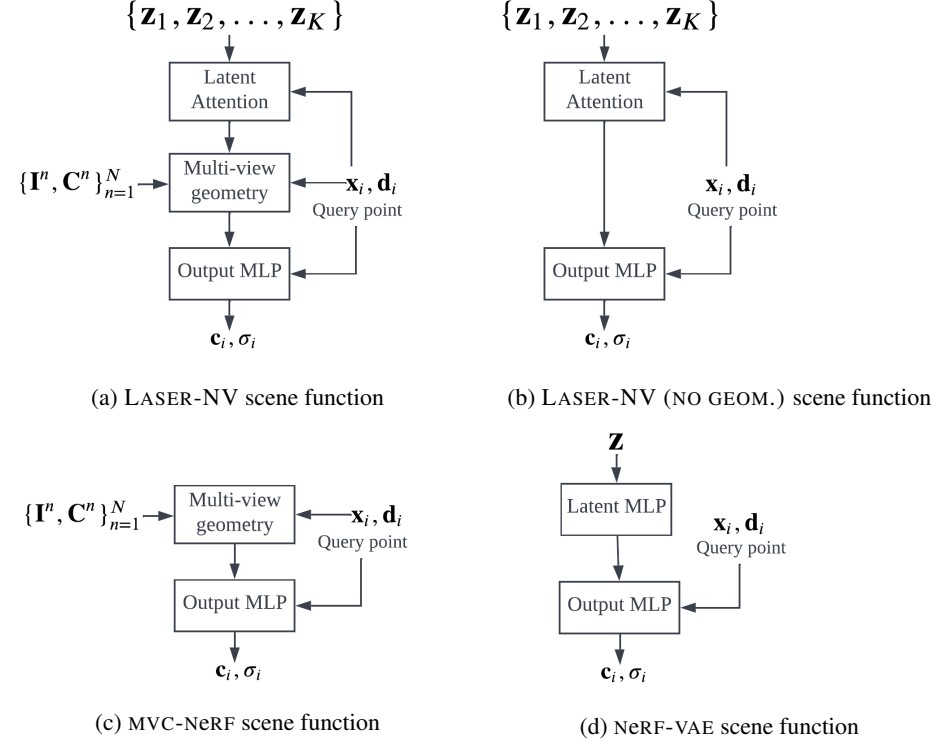

(a) LASER-NV scene function

(b) LASER-NV (NO GEOM.) scene function

(c) MVC-NERF scene function

(d) NeRF-VAE scene function

Figure 7: Scene function architecture for LASER-NV (a), its ablations (b) and (c), and NeRF-VAE (d).

**Latent attention** A query point $x_i$ attends to $\mathcal{Z}$ as per Fig. 7 (a) using a cross-attn$(\gamma(x_i), \mathcal{Z}, L)$, where $\gamma(x_i)$ is the circular encoding function of positions (see Appendix A.4). We use the same architecture across datasets, setting $L = 3$ layers, 512 hidden units, and 4 attention heads.

**Local features from multi-view geometry** We follow PixelNERF's design of stacking the features of the encoded images at different levels of resolution, which allows the conditioning features to

capture context structure at different scales. For input images of resolution $W \times H$, we stack the features of the outputs from all layers after the first layer:

$$\mathbf{H}^n = \text{stack}(\{\mathbf{H}_l^n \in \mathbb{R}^{\frac{W}{i} \times \frac{H}{i} \times 16i}\}_{l=1}^{L}), \tag{9}$$

where $i = 2^l$. When stacking, we upscale the feature maps of all the feature maps of layers $l \in \{2 \dots L\}$ to the size of the feature map with largest resolution ($l = 1$) using bilinear interpolation. The resulting set of stacked feature maps (one for each context view) is then used to condition the scene function.

As we described in Section 3.2, our multi-view geometry module is in part based on pixelNeRF's processing of context views. Particularly, we process local features with exactly the $f_1$ ResNet detailed in Yu et al. (2021): positions and directions are projected to the view-space using the cameras intrinsic parameters, and then processed with 3 residual blocks of 512 hidden units and ReLu activation. When computing the projection, we also define the binary variable $v_i^n$ based on whether the projected coordinates are within the image plane.

The output features are integrated with our latent features via $M_\theta$ which is a cross-attention module. Note that context features with $v_i^n = 0$ are not attended to (we modify the pre-softmax logits of those context features by adding a large negative number to the logits). In our experiments $M_\theta$ has 3 layers and 512 hidden units.

**Scene function output MLP** We now describe the final component of the scene functions of all models, as shown in Fig. 7. This component follows the residual MLP architecture of the scene function of NeRF-VAE. Namely, it takes as input the circular encoded positions $\gamma(\boldsymbol{x}_i)$ and directions $\gamma(\boldsymbol{d}_i)$, and at every layer it linearly projects the conditioning features (i.e. $\boldsymbol{z}$ for NeRF-VAE, or $\boldsymbol{f}_i$ for LASER-NV) and sums it to the activations. The MLP is composed of two parts: the first has 4 layer of 256 hidden units each and outputs the densities, and the second part has an additional 4 layers and outputs the colours. All layers use a swish activation function.

The circular encoding function $\gamma$ is given by

$$\gamma(p) = (\sin(2^{L_{\min}}\pi p), \cos(2^{L_{\min}}\pi p), \dots, \sin(2^{L_{\max}}\pi p), \cos(2^{L_{\max}})) \tag{10}$$

where we use different constants $L_{\min}$ and $L_{\max}$ for positions and directions and for each dataset (listed in Appendix E).

### A.5 NeRF-VAE and conditional NeRF-VAE

Our implementation NeRF-VAE follows the same overall architecture as that of Kosiorek et al. (2021), but we use the image encoder described in Fig. 6 instead. The approximate posterior is computed as a diagonal Gaussian distribution, the means and log-variances of which are given by an MLP $e$ as a function of the pooled feature maps $\mathcal{H}$ (pooled with a simple averaging over feature maps). The MLP $e$ has two layers of 256 hidden units and ReLu activation function. The scene function is composed of an MLP that processes the latent vector $\boldsymbol{z}$ (one linear layer of 256 units, followed by a ReLu non-linearity), followed by the output MLP described in Appendix A.4.

### A.6 Background scene function

The volumetric rendering integral of a ray $\boldsymbol{r}$ is given by

$$\boldsymbol{c}(\boldsymbol{r})_{\text{NeRF}} = \int_{t_n}^{t_f} T(t)\sigma(\boldsymbol{r}(t))\boldsymbol{c}(\boldsymbol{r}(t), \boldsymbol{d}) \, dt \,, \text{ with } T(t) = \exp\left(-\int_{t_n}^{t_f} \sigma(\boldsymbol{r}(s)) \, ds\right). \tag{11}$$

If the ray is infinite (camera's far plane $t_f \to \infty$) the colour weights $T(t)\sigma(\boldsymbol{r}(t))$ would always sum to one. For a truncated ray (finite $t_f$), this happens only if the ray passes through a solid surface. This heuristic allows to model distant backgrounds by putting them at the end of the integration range, but it doesn't work if the camera position changes significantly compared to the ray range $t_f - t_n$. If not normalized, the final weight $T(t_f)$ tell us about the probability of light hitting a particle beyond the ray.

In that case, we can model the background as an infinite dome around the scene and model it with a light-field that depends only on the camera direction. That is, we compute the colour as

$$\boldsymbol{c}(\boldsymbol{r}) = \boldsymbol{c}(\boldsymbol{r})_{\text{NeRF}} + T(t_f)f_{bg}(\boldsymbol{d}) \,. \tag{12}$$

For the City dataset, in order to model the sky and sun, $f_{bg}(\mathcal{Z}, \boldsymbol{d_i})$ is a learned neural network. We use a cross-attention module with 2 layers to attend to $\mathcal{Z}$. For NeRF-VAE and cond. NeRF-VAE we instead use a 2 layered MLP with ReLu activations. For the ShapeNet $f_{bg}$ is a constant colour (white).

We address the challenging backgrounds of MSN-Hard with a more expressive attentive background scene function that conditions on both the latents $\mathcal{Z}$ as well as context image features – this is in a similar fashion to how the NeRF scene function of LASER-NV integrates its latents with attention to local context features. Given an encoded ray direction $\gamma(\boldsymbol{d}_i)$ and context feature maps $\bigcup_{n=1}^{N} \boldsymbol{H}^n$, a cross-attention module attends to the latents: $\hat{\boldsymbol{h}}_i^{bg} = f_{bg}^1(\gamma(\boldsymbol{d}_i), \mathcal{Z})$, and the output is used to attend to the set of context features $\bigcup_{n=1}^{N} \boldsymbol{H}^n$ with a second cross-attention module $\boldsymbol{h}_i^{bg} = f_{bg}^2(\hat{\boldsymbol{h}}_i^{bg}, \bigcup_{n=1}^{N} \boldsymbol{H}^n)$. Finally, an MLP $f_{bg}^{mlp}$ computes the background colour by taking as input the obtained background features and additionally conditions on the encoded directions by adding a linear projection of the directions to the activations of every layer. This MLP is composed of 2 layers of 512 hidden units each with a ReLu nonlinearity, and outputs the RGB background colours using a sigmoid nonlinearity.

## B CITY DATASET

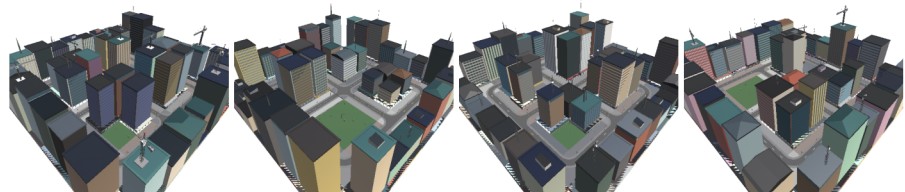

Figure 8: Examples of generated cities.

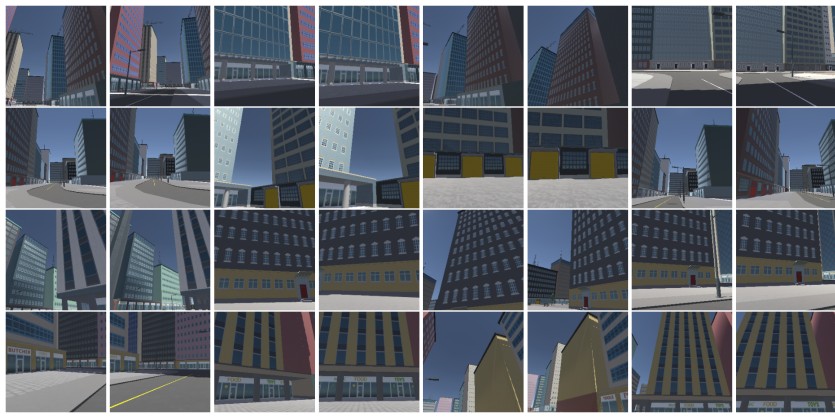

Figure 9: Examples of different viewpoints (column) for each scene (row).

The aim of the City dataset is to allow generating visually complex scenes with large variability in its structure and appearance. Each generated city consists of a large area spanning approximately $270m \times 270m$, which consists of a 2x2 grid of blocks, where each block is composed of 4 lots. A lot can be a building or a green patch (park). The city 2x2 grid is surrounded by an outer ring of buildings. Roads separate all blocks in the scene. Each building is randomly generated with variation in size, appearance (textures, materials, colours), architectural styles (residential, industrial, commercial). Fig. 8 shows examples of generated city scenes. Once a city is built, we specify a range of points of interest, i.e. road intersections and parks and use each of these points as the starting placement for a *scene* (data point) of our dataset. Each scene is comprised of posed images rendered from randomly

chosen viewpoints near the point of interest. More specifically, we render 8 randomly chosen nearby points, and 4 random viewpoints for each point, making a total of 32 generated views per scene. Each generated view is annotated with the ground-truth render, camera parameters and depth maps. Our generated training dataset contains 100,000 scenes generated in total from approximately 2,600 uniquely generated cities. We test on 3,200 scenes generated in total from 10 uniquely generated cities.

## C  ADDITIONAL VISUALIZATIONS

**Capturing detail at different scale**   In Fig. 10 we can see at a higher resolution how our model is expressive enough to capture texture and geometry details at different scales (e.g. far-away buildings and nearby objects such as the lamp post and the colonnade).

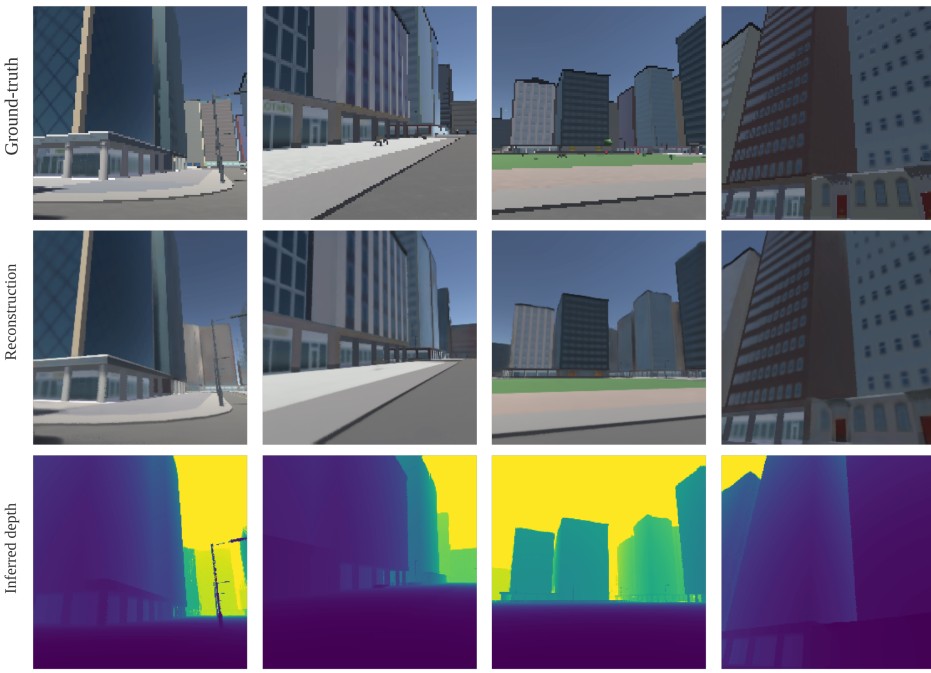

Figure 10: Higher resolution reconstructions of LASER-NV reveal detail captured at different scales (e.g. lamp posts, and nearby buildings vs buildings at a large distance).

**Novel view synthesis with ShapeNet NMR**   In Fig. 11 we compare predictions for a number of objects that reveal how NeRF-VAE produces blurry reconstructions compared to MVC-NeRF and LASER-NV. When an image does not reveal the full shape of an object, as shown in Fig. 12, LASER-NV is able to generate multiple plausible variations (in contrast to a deterministic which generates a single blurry prediction).

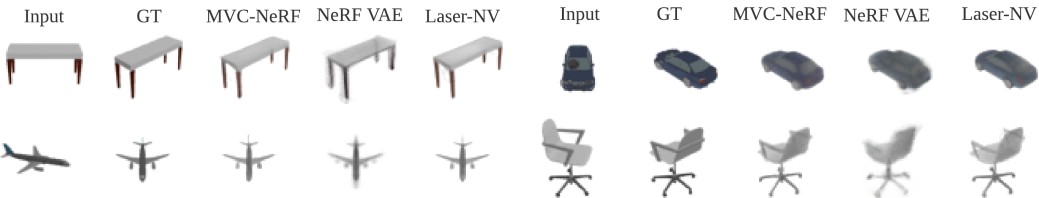

Figure 11: Samples of category-agnostic ShapeNet for ablations of LASER-NV. MVC-NeRF and LASER-NV both result in high-fidelity predictions, while the NeRF-VAE's outputs are blurrier.

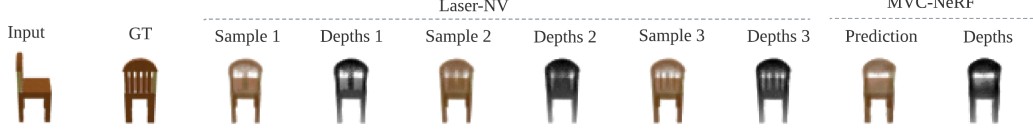

Figure 12: When predicting novel view from an ambiguous input, LASER-NV generates plausible variations, whereas MVC-NERF results in a single blurry prediction.

**MSN-Hard predictions**    Figure 13 shows LASER-NV's predictions on MSN-Hard including depth maps estimated from NERF's densities.

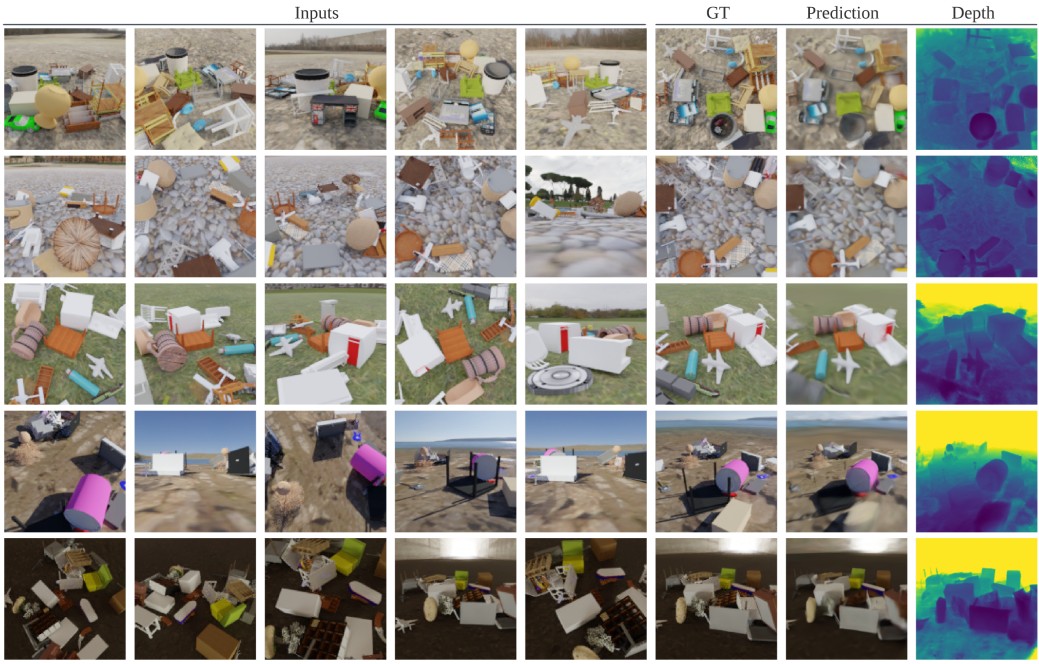

Figure 13: Representative examples of LASER-NV predictions on MSN-Hard.

## D    ADDITIONAL RESULTS

**Data efficiency**    We evaluate how data hungry LASER-NV is compared to NERF-VAE and MVC-NERF. In Figure 14 we report the reconstruction colour log-probability estimated on a validation set of the City dataset. For LASER-NV, we observe little degradation between using 50K scenes and 12.5K unique scenes. For NERF-VAE we notice a larger drop when reducing the size from 25K to 12.5 scenes, showing clearer signs of overfitting. Finally, since MVC-NERF severely underfits the City dataset for all training set sizes, we do not see any clear performance drop.

**Test-Time Scaling of the Latent Set**    We further want to understand how different latent set sizes at *test time* impact performance. Figure 15 shows reconstruction performance for different values of $K$ at test time, for various values of used $K$ during training. As we can see, LASER-NV does not significantly leverage a larger latent set size than trained with. As expected, we do see performance degrading when using *smaller* sets at test time, and this effect becomes stronger for models trained with larger capacity.

**Training LASER-NV without ground-truth depth**    We train all our models using ground-truth depth on the City dataset. This reduces memory and computation requirements during training. We

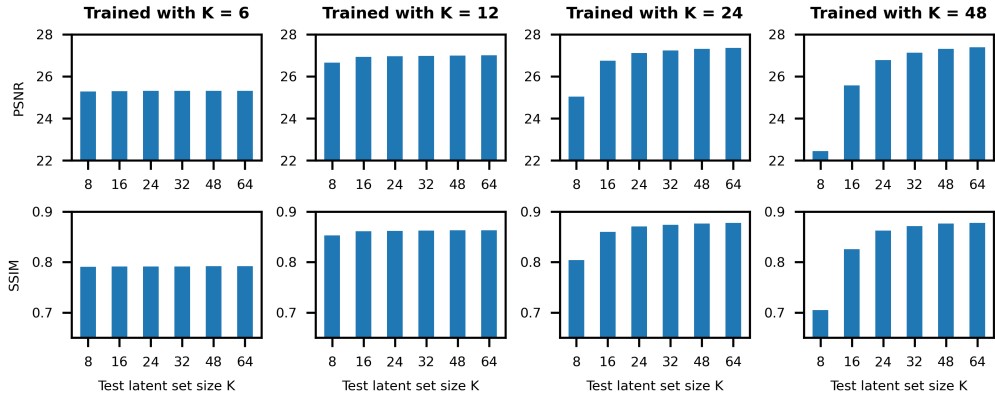

Figure 14: Validation reconstruction performance of different method as a function of number of scenes throughout training.

Figure 15: Reconstruction performance of LASER-NV as a function of the latent set size used at test time for different models trained with increasing sizes $K$.

note, however, that it is also possible to train LASER-NV without using ground truth depths. Once trained, we find that both the model trained with depths and the model trained without achieves slightly higher average colour log-likelihood (3.87) compared to LASER-NV (of 3.83) (importance-weighted estimate using 10 samples). We hypothesize that this is because the model trained without depths is trained solely using the rendering method used at test time.

As any NeRF-related method, LASER-NV can output estimated depth maps easily. We show example reconstructions and *inferred* depth maps of a trained LASER-NV on the City in Fig. 16. Note how the model is capable of capturing fine detail in textures and shapes (e.g. lamp post).

**Computational efficiency of LASER-NV vs NeRF-VAE** For a model conditioned on 10 input views, inference (inferring the latent distribution) and rendering of a single image for LASER-NV (with 24 slots) take 0.39 GFLOPs and 1.80 GFLOPs respectively. NeRF-VAE instead takes 0.37 GFLOPs and 0.15 GFLOPs. The main reason for more expensive rendering in LASER-NV is the introduction of PixelNeRF-like conditioning. A model with a flow posterior but without such conditioning uses 0.47 GFLOPs to render a single image. Attention, however expensive, is evaluated only once per scene (to compute the prior/posterior). The PixelNeRF features are cheap per point, but they are evaluated for every 3D point used in rendering, therefore dominating the overall cost. Note that the above numbers do not reflect training latency. At training time we can guide the rendering process by using depths, which allows us to evaluate the scene function only twice for every pixel. We also

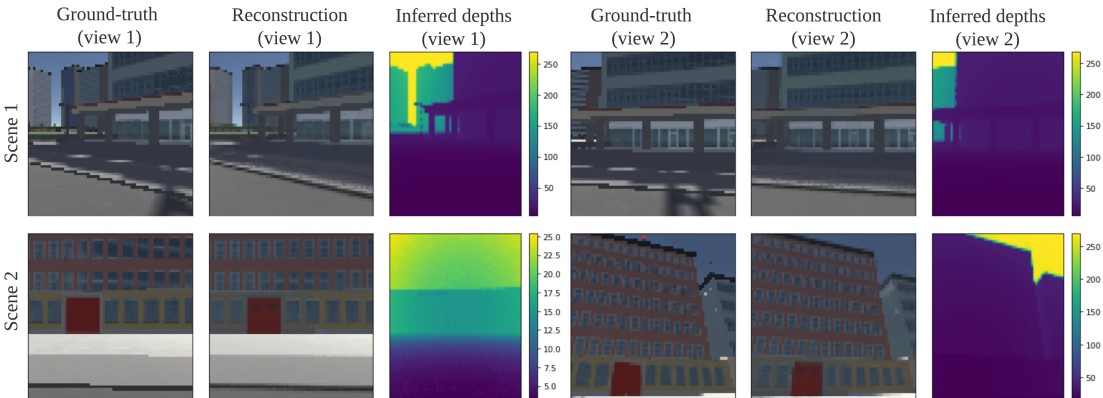

| Ground-truth (view 1) | Reconstruction (view 1) | Inferred depths (view 1) | Ground-truth (view 2) | Reconstruction (view 2) | Inferred depths (view 2) |

Figure 16: Reconstructions of LASER-NV trained without the use of ground-truth depths.

subsample images and we never reconstruct the whole target view(s). This makes training faster, and means that LASER-NV is only about 4x slower per training iteration than NeRF-VAE. Having said that, we noticed that NeRF-VAE does not benefit from longer training (its performance saturates), while the performance increase from increasing the model size (to equalize the FLOPs) is negligible.

**Robustness to camera noise** We carry out an experiment on MSN-Hard in which we train and test with camera poses with incremental levels of Gaussian noise. We replicate the set up in Sajjadi et al. (2021) and compare their reported results using SRT and PixelNeRF to LASER-NV. Results in Fig. 17 show that while LASER-NV outperforms the other models when there is no noise, its performance drops more quickly compared to SRT, in a similar fashion to PixelNeRF. However, LASER-NV is more robust than PixelNeRF for all levels of noise. We believe that the use of NeRF and local features makes these models less robust to noisy cameras. That said, there are recent methods that allow mitigating noise sensitivity with NeRF (Lin et al., 2021) that can be incorporated.

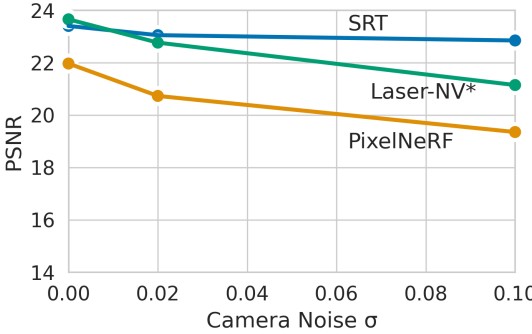

Figure 17: PSNR of models trained and evaluated under cameras with different levels of noise. The PSNR values of LASER-NV* are preliminary training values models that have not trained to convergence (340K steps out of 1000K).

## E  TRAINING DETAILS

**Hierarchical rendering** Instead of optimizing separate sets of parameters for the coarse and fine rendering passes, we find that it suffices to only keep separate parameters of $F_\theta$. This allows sharing parameters (and reduce computation) when querying the scene function on the coarse sampled points with respect to latent attention and multi-view geometry components of models.

**Training with depth (City)** Because we have RGB-D data in the City dataset, we can use depth as a supervised signal following the method in Stelzner et al. (2021). This method involves replacing the

pixel colour likelihood, for a given ray $r$, with the ray colour and depth likelihood $p(\mathbf{t}(r), \mathbf{c}(r) \mid \mathcal{V}, \mathcal{Z})$, where $\mathbf{t}(\mathbf{r})$ is the ray's ground-truth depth. This alternative likelihood requires only two scene function evaluations per ray. In practice, we find that we get the best quality renders when combining both log-likelihood terms to the loss (each weighted by 0.5). At every training set, similar to how NeRF-VAE is trained, the likelihood terms are estimated by only reconstructing a subset of rays (i.e. pixels) of all the target images. We find that using 64 rays for estimating the (expensive) image log-likelihood, and 512 to estimate the ray colour and depth likelihood term, works well in the City. Additionally, we restrict output densities to be between 0 and 10 by using a weighted sigmoid in the output of the scene function.

**Additional regularization**   The loss function for ShapeNet experiments includes a density regularization term based on the L1-norm of the densities, with a weight of $0.01$. For all experiments, we scale the model parameter updates (Pascanu et al., 2013, Sec. 3.2), such that loss gradient norms have a maximum value of 10 for improved stability.

**Hyper-parameters**   We use the following hyper-parameters across all models. Some of which were selected for each dataset based on a preliminary exploration using the training dataset and a validation split.

| Hyperparameter | City | Shapenet NMR | MSN-Hard |
|---|---|---|---|
| Learning rate | 0.0003 | 0.0002 | 0.0001 |
| Batch size | 96 | 96 | 32 |
| # of rays per scene | 512 | 512 | 384 |
| # of context views | 2 | 1 | 5 |
| # of target views | 8 | 23 | 3 |
| # of posterior context views | 10 | 4 | 8 |
| # of coarse points | 256 | 32 | 64 |
| # of fine points | 64 | 64 | 32 |
| Camera $t_{near}$ | 0.05 | 0.1 | 0.01 |
| Camera $t_{far}$ | 270 | 3.7 | 19 |
| Positions circular encoding $L_{\min}$ | -8 | -2 | -5 |
| Positions circular encoding $L_{\max}$ | 8 | 8 | 10 |
| Directions circular encoding $L_{\min}$ | 0 | 0 | 0 |
| Directions circular encoding $L_{\max}$ | 4 | 4 | 8 |
| LASER-NV set size K | 24 | 8 | 32 |
| LASER-NV latent vector dimensionality | 128 | 128 | 128 |

Table 4: Common hyper-parameters

