# OpenReview forum: "Laser: Latent Set Representations for 3D Generative Modeling"
_ICLR.cc/2023/Conference — Submitted to ICLR 2023_

### Official Review · Reviewer_kWNb · 2022-10-17

**Confidence:** 3
**Correctness:** 3
**Technical Novelty And Significance:** 3
**Empirical Novelty And Significance:** 3
**Recommendation:** 5

**Clarity, Quality, Novelty And Reproducibility:**

Clarity: As mentioned above, some mathematical equations should be included to make it clear.

Quality: The result outperforms the baseline methods on synthetic data. However, it is unclear how it performs on real data.

Novelty: The latent set representation is novel.

Reproducibility: It could be reproducible if the clarity problem is resolved.

**Strength And Weaknesses:**

+ The idea of using the latent set representation is novel.

+ The design of multiple latent variables makes it possible to make multiple different predictions in unobserved regions.

+ LASER outperforms MVC-NeRF and NeRF VAE in experimental analysis. LASER produces a sharper result.

- The experiments are done on synthetic data only, where the camera pose is accurate. However, in real cases, there is always error in camera poses. It would be good if the author could show the result on real data (like in the original NeRF paper) so that we could understand its robustness to pose errors.

- The explanation on obtaining "z" for rendering is unclear. It would be good if mathematical equations are provided in Sec. 3.3 "Querying the Latent Set" so that the readers could better understand how "z" is computed and where the randomness comes from.

- Although the results are better than baseline results, they are still blurry, e.g., in Fig. 3. It should be analyzed as limitations. However, the paper did not analyze the limitations of the proposed method.

**Summary Of The Paper:**

This paper proposes a latent set representation for 3D generative modeling. The proposed method includes a normalizing flow based context encoder, a NeRF scene function, and a VAE based renderer. By using a set of latent variables rather than one latent variable, the proposed method can make multiple different predictions for unobserved regions in novel view synthesis. The experiments show that the proposed method outperforms baseline methods including MVC-NeRF and NeRF VAE.

**Summary Of The Review:**

The paper is novel. However, the experiments on real data are missing. So it is unclear how sensitive it is to camera poses. Thus I'm giving the borderline reject rating. I would like to change my rating if my concern listed above is well addressed.

---

> ### Author Response · Authors · 2022-11-16
> **Response**
>
> We thank the reviewer for their feedback and refer to our shared response that addresses the lack of experiments on real data and robustness to pose errors.
>
> ## Noisy cameras
> To understand how Laser-NV behaves with noisy cameras, we replicated the noisy camera study from SRT on the MSN dataset (see Appendix D): We trained the model by using noisy input cameras (with Gaussian noise in the SE(3) space). The model performance degrades with increasing noise levels, very similarly to PixelNeRF. This is expected given that Laser-NV uses PixelNeRF’s conditioning mechanism which is sensitive to known cameras. However, we note that:
>
> * Our model remains significantly better than PixelNeRF, even at high noise levels.\
> * There exist a number of techniques in the literature that can be used to reduce the noise level in preprocessing (e.g. COLMAP [2]) or be integrated into the training procedure (e.g. BARF [3])
> * Sensitivity to camera noise is shared across all NeRF based models and not unique to Laser-NV.
>
> Finally, we note that the models we train with noisy cameras are not yet converged (350k training iterations vs 1M for a fully-trained model), and we expect the performance gap to further decrease.
>
> #### References
> 2. Schoenberger et. al. Structure-from-Motion Revisited. CVPR 2016.
> 3. Chen-Hsuan Lin et al. BARF: Bundle-Adjusting Neural Radiance Fields. ICCV 2021
>
>
> ## How the Latents are Obtained
> We have expanded Sec. 3.1 to include the equation that defines the distribution of the latents, and how they are generated starting from a base distribution (Gaussian) and passed through a sequence of invertible mappings parameterized by transformers.
>
> ## Limitations
>
> While we hope to have addressed the concerns on lack of real data in our shared response, we have included a comment in the conclusions on that regard.

---

> > ### Comment · Reviewer_kWNb · 2022-12-02
> > **Response**
> >
> > Thanks for the clarification. The worry I have is that the synthetic noise and real noise might have different patterns (including camera pose noise, and pixel noise). Thus a good performance on synthetic datasets does not mean it also works in real scenes. Just want to know what the blocker is for showing real-world results in the paper.

---

> > > ### Author Response · Authors · 2022-12-02
> > > **Not Enough Data for Real World Results**
> > >
> > > Thank you for clarifying your concerns. The main blocker for showing real-world results in our paper is the lack of a suitable dataset. Our model requires a dataset of static 3D scenes with several views per scene. In our early experiments we noticed that LASER-NV overfits significantly if we use 10k scenes. It seems that using 100k scenes eliminates any overfitting issues. We are not aware of any real-world dataset with these properties (static, several views per scene, >10k scenes), but if you do know any, we would welcome any pointers. Please note that amortized NeRF-like models have been previously published without real-world results, e.g. Sajjadi et. al., "Object Scene Representation Transformer", NeurIPS 2022.

---

> > > > ### Comment · Reviewer_kWNb · 2022-12-02
> > > > **Response**
> > > >
> > > > Thanks for the explanation. Does the google Street View dataset meet your requirement? It can be downloaded from google street view website with api. It has been used in the following paper: Scene Representation Transformer:
> > > > Geometry-Free Novel View Synthesis Through Set-Latent Scene Representations

---

> > > > > ### Author Response · Authors · 2022-12-02
> > > > > **StreetView**
> > > > >
> > > > > Thanks for the suggestion. While this dataset might be big enough and has multiple views per scene, it is not static. As you can see on the [SRT's project website](https://srt-paper.github.io/data/streetview.html), there are objects that move between frames (e.g. cars). In order to support this dataset SRT had to incorporate tricks from [NeRF in the Wild](https://arxiv.org/abs/2008.02268) to explain transient (moving) elements. While it is possible to use similar tricks with our model, they would require appropriate probabilistic treatment to stay compliant with the VAE framework, which is not trivial.
> > > > >
> > > > > It would certainly be interesting to extend LASER-NV to dynamic scenes, but this is beyond the scope of the current paper.

---

> > > > > > ### Comment · Reviewer_kWNb · 2022-12-12
> > > > > > **Response**
> > > > > >
> > > > > > Thanks for the reply. I still suggest add the experimental results on real datasets like street view, even if it fails. It would help understand the limitation of the method (i.e., it is hard to adapt the current method to more challenging scenes).

---

> > > > > > > ### Author Response · Authors · 2022-12-12
> > > > > > > **Real world data**
> > > > > > >
> > > > > > > We thank the reviewer for the suggestion to experiment on the StreetView dataset, we agree it can be useful to the reader to know the limits of the model on real world data. We will thus aim to train Laser-NV on it and provide results in the final version, even if to show qualitative results.
> > > > > > >
> > > > > > > That said, during the rebuttal we have carried out experiments to address the reviewer's concerns:
> > > > > > > - We've included results on the challenging MSN-Hard experiment and show that Laser-NV makes high fidelity predictions of pixels and 3D structure.
> > > > > > > - We've further updated the paper with experiments with artificial noise to the camera poses to simulate errors in camera, and have shown that Laser-NV still outperforms most other methods (except for SRT) despite its degraded performance.
> > > > > > >
> > > > > > > We would like to conclude by noting that novel view synthesis from very sparse views is a largely unsolved problem. This is even more the case for 3D scene generative models which are capable of making predictions under partial observability. This capability comes at the cost of requiring larger static and posed datasets, something that is still scarce.
> > > > > > > We believe, and hope the reviewer agrees with us, that novel methods that significantly push the capabilities of these methods in reasonably complex synthetic datasets (i.e. increasingly closer to real-world scenes) are worthwhile contributions to the community.

---

### Official Review · Reviewer_b7vH · 2022-10-25

**Confidence:** 3
**Correctness:** 3
**Technical Novelty And Significance:** 2
**Empirical Novelty And Significance:** 2
**Recommendation:** 5

**Clarity, Quality, Novelty And Reproducibility:**

The paper is clear and the experiments are detailed. I think it is reproducible.

**Strength And Weaknesses:**

Strengths:
- The proposed latent set representation is useful. According to the experiments, the latent set representation has a good scaling ability.
- The proposed methods have a better performance in predicting unobserved views compared to NeRF-VAE.

Weaknesses:
- The current City dataset is relatively simple. Can the method work on more complex scenes, such as multiple ShapeNet objects (like PixelNeRF) or real dataset?
- I am concerned about the technical novelty of this paper. The idea of set latent scene representation is actually not a new thing, like Scene Representation Transformer, CVPR 2022. Moreover, aggregating multi-view local features is a standard technique in computer vision, and has been widely used in many existing works, including PixelNeRF, IBRNet etc. The proposed frameworks seem to be a combination of existing techniques in spite of the performance improvement. I am afraid the novelty is below the bar of ICLR.


**Summary Of The Paper:**

The paper presents a conditional generative model based on NeRF-VAE. The differences are that the proposed methods use a set of latent vectors to encode the scenes instead of one and it incorporates multi-view geometry local features for better feature learning. It also applies normalizing flow to model conditional prior and conditional posterior. Experiments show that the method outperforms baselines on both reconstructions of observed views and predictions of unobserved novel views.

**Summary Of The Review:**

 I am concerned about the novelty of this paper in spite of the performance improvement. I think this paper is below the bar of ICLR.

---

> ### Author Response · Authors · 2022-11-16
> **Response**
>
> We thank the reviewer for their valuable feedback and refer to our shared response. We hope it addresses the main concerns on lack of evaluation on more challenging scenes, as well as regarding the novelty of the paper in relation to SRT.
>
> We kindly ask the reviewer which claims are not well supported so we can address them.

---

> > ### Comment · Reviewer_b7vH · 2022-11-28
> > **Response**
> >
> > Thanks for the clarification. The new experiments address my first concerns. I would like to change my scores from 5 to 6. However, I am still concerned about the novelty of the proposed latent set representations.

---

> > > ### Author Response · Authors · 2022-12-02
> > > **Unclear Concerns Regarding Novelty**
> > >
> > > Thank you for increasing your score. Just to clarify, we do not claim that the set representation in our paper is novel. The novelty is in using said representation as the latent representation of a VAE, applying it to 3D scene modelling, and characterizing its benefits (better scaling properties, higher expressivity). The model resulting from this combination has not been described before.
> > >
> > > With the above in mind, would you mind sharing what exactly are you concerned about regarding novelty?

---

### Official Review · Reviewer_k538 · 2022-10-25

**Confidence:** 3
**Correctness:** 4
**Technical Novelty And Significance:** 3
**Empirical Novelty And Significance:** 2
**Recommendation:** 6

**Clarity, Quality, Novelty And Reproducibility:**

The paper is clear and easy to follow.  Combining permutation-invariant normalizing-flow and transformer is novel and interesting.

Transformer is popular and also used in other novel view synthesis papers such as IBRNet.
There are existing work combining generative modeling (GAN/VAE/diffusion-process) with NERF, such as NERF-VAE, GRAF, SGN, ...
Representing a scene by piecewise implicit functions, grids or octrees has also been proposed before, which is related to the proposed set-based reprsentation.

**Strength And Weaknesses:**

+ The  permutation-invariant set normalizing flow is an interesting idea, which naturally enables representation learning with transformers.

Some questions:
Is there any physical meaning for latent codes z? are they different objects? or different position?
There are existing work combining generative modeling (GAN/VAE/diffusion-process) with NERF, any insight why the proposed method is better?

**Summary Of The Paper:**

The paper proposed a VAE-NERF model for novel view synthesis:the latent code z is represented by a permutation invariant set and conditional prior p(z|v) as a permutation-invariant normalizing flows; the query k/v pairs are modeled from latent z to decode the scene and images. The results look plausible and unobserved parts can also be generated.

**Summary Of The Review:**

The paper proposed a VAE-NERF model for novel view synthesis:
a latent set z is learned by a permutation invariant set and conditional prior p(z|v) as a permutation-invariant normalizing flows; the query k/v pairs are modeled from latent z to decode the scene and images. The results look plausible and unobserved parts can also be generated.

The idea is interesting and novel, naturally combining normalizing flow to model NERF-VAE. The experimental result looks satisfactory in synthetic single objects and urban scenes.

---

> ### Author Response · Authors · 2022-11-16
> **Response**
>
> We thank the reviewer for their comment and refer to the shared response regarding the interpretation of the latent code.
>
> ## Why is LASER-NV better than existing GAN, VAE and diffusion-models?
>
> * GANs: Compared to GANs, we provide an (efficient) inference mechanism which allows for novel view synthesis. VAEs also allow estimating the log likelihood of the data, and do not suffer mode collapse as GANs do.
> * VAEs: We provide much better performance (reconstruction PSNR, novel view synthesis, uncertainty estimation) than existing NeRF-based VAEs which use a simplistic latent structure and do not leverage local features attended via multi-view geometry.
> * Diffusion models: We provide latent scene representation that can be used in downstream tasks. We are also not aware of published diffusion scene models that can do novel view synthesis.

---

### Official Review · Reviewer_TTqh · 2022-10-25

**Confidence:** 4
**Correctness:** 3
**Technical Novelty And Significance:** 2
**Empirical Novelty And Significance:** 2
**Recommendation:** 6

**Clarity, Quality, Novelty And Reproducibility:**

- Clarity

The paper is clearly written and easy to understand and the figures make the paper easier to understand.


- Novelty

The work seems to be a mixture of NeRF-VAE and SRT which is my biggest concern. Although the taking advantages of other methods does not necessarily reduce the novelty of the work, using the latent code of NeRF-VAE, thereby having multimodality and using set-representation latent code of SRT is straightforward with no huge novelty in the problem setting.

- Quality

The experimental design of the paper is well, although some more ablation studies could help.

- Reproducibility

Code is not uploaded. Wil the code and city datasets be uploaded? I think the city dataset will be used alot in the community.


**Strength And Weaknesses:**

**Strength**
- Performance

By looking at the figures and tables of the manuscript, the work clearly outperforms NeRF-VAE [1], producing less blurry images from multiple views. However, I would like to ask the authors why SRT [2] was not used as a baseline.

- Good experimental design

I enjoyed showing that increasing the latent vector size of NeRF-VAE does not result in a good performance, thereby the design choice of this work is better. The ablations by showing with performance with/without either latent set and local features make the paper stronger.


**Weaknesses**
- Novelty

My main concern lies in the novelty of the work. The work seems to be a mixture of NeRF-VAE and SRT. Please see the novelty section below for details.


- Intuition behind the set data structure for the scene latents

Although it is clear from Table 1, that the scene latents increase the performance of the method, I would like to ask the authors the intuition behind the sets as the choice for the latent data structure. Other works such as Neural Sparse Voxel Fields [3] either have an explicit latent data structure, where the latent code explicitly encodes local information or alot of other generative modeling works show the controlability of latent code by interpolating them. I do not see either of the intuition behind the set representation nor other attributes of set-valued latent representation and would like to hear from the authors the reason for choosing particularly this data structure besides the performance.

- Proposed model requires depth information

Although the authors have reported the log-likelihood score of the method in the appendix without depth information, reporting all the scores in Table 1 without depth information will strengthen the paper.

[1] Kosiorek et al. NeRF-VAE: A Geometry Aware 3D Scene Generative Model. ICML, 2021

[2] Sajjadi et al. Scene Representation Transformer: Geometry-Free Novel View Synthesis Through Set-Latent Scene Representations. CVPR, 2022

[3] Liu et al. Neural Sparse Voxel Fields. NeurIPS, 2020


**Summary Of The Paper:**

This paper tackles the problem of scene generation, generating consistent images of scenes from multiple viewpoints conditioned on few images. The main technical contribution of the work is proposing a set-valued latent representation using normalizing flows, built on top of NeRF-VAE. The proposed model is tested on synthetic city dataset and ShapeNet dataset, while having competitive performance with respect to recent methods.

**Summary Of The Review:**

My biggest concern lies in the novelty of the method. I would like to hear from other reviewers before making the final decision.



---------------------------
After Rebuttal:
I appreciate the authors for their response.
I'm in favor of accepting the paper, due to it's multimodality and high performance.
I have raised my score accordingly.

---

> ### Author Response · Authors · 2022-11-16
> **Response**
>
> We thank the reviewer for their feedback and refer to our shared response to common reviewer questions or concerns, including a comparison with SRT and the intuition behind the latent structure.
>
> ## Proposed model requires depth information
>
> None of the models in our experiments require depth information. The use of depth information is used to train the models in a computationally more efficient way, thus reducing research and energy costs. As we show in the appendix, our model can render equally high-fidelity results when trained with or without depth, with very similar test marginal likelihoods. Because all the models are trained in the same way, and evaluated without using depths, results are fully comparable.
>
> ## Will the code and data be uploaded?
>
> We agree that the City can be a useful dataset to the community and are going to release it with the final version of the paper. While we don't plan on releasing the code, we will include a detailed pseudo code of the model in the appendix.

---

> > ### Comment · Reviewer_TTqh · 2022-11-25
> > **Regarding depth**
> >
> > I appreciate the author's response.
> >
> > Still, I'm a bit confused using depth as input information.
> > In section 4.4, the manuscript says
> >
> > >For experiments with the City, we leverage
> > ground-truth depth maps using the method proposed by Stelzner et al. (2021) in order to train all the
> > models with fewer scene function evaluations per ray. Not using ground-truth depth significantly
> > increases training time while only marginally lowering reconstruction PSNR, see Appendix D for a
> > discussion and Fig. 16 in Appendix B for visualisations.
> >
> > This sounds to me that the model is using ground-truth depth maps for training.
> > Could the authors clarify this statement?

---

> > > ### Author Response · Authors · 2022-11-29
> > > **Depth use clarification**
> > >
> > > In City experiments, we make use of an unbiased formulation of volumetric rendering as the training objective [1], which allows for more efficient training when depth information is available. In no case is depth information used as *input* to the model, neither during training nor testing, it's only used in the loss.
> > > We would also add that:
> > > - Experiments on City use the above training technique for efficiency, but we emphasize that it is not a requirement in any way. This is demonstrated by the comparable results of training Laser-NV on City without depth-based supervision (appendix D). Because Laser-NV performs comparably in both cases, it thus outperforms all other models even when trained with or without depth supervision.
> > > - ShapeNet and MSN-Hard datasets do not have depth maps (thus depth map supervision is not used in those results). Quantitive results show further evidence that Laser-NV does not require depth maps, also evidenced by the high fidelity of colours and 3D densities (Fig 3 and appendix C).
> > >
> > > For the above reasons we strongly believe our reported results are accurate and representative of the performance of our models.
> > > That said, we agree that reporting results of all models shown in Table 1. without using depth supervision can be informative to the reader, and we will provide those numbers in the final version.
> > >
> > > Let us know if this addresses your concern.

---

> > > > ### Comment · Reviewer_TTqh · 2022-12-01
> > > > **Regarding depth**
> > > >
> > > > I appreciate the authors response for the answer.
> > > >
> > > > If the depth information is used in the loss function, it means that the method is supervised with the depth information.
> > > > This may result in unfair comparison when comparing with the baseline regardless of if it is used as the input or not. I would appreciate if the authors provide the full results of Table 1.

---

> > > > > ### Author Response · Authors · 2022-12-01
> > > > > **Depth use clarification**
> > > > >
> > > > > We thank the reviewer for their response.
> > > > >
> > > > > We fully agree that if the baselines were trained without depth supervision, results would not be comparable against Laser-NV. However, we ensured that all City models and baselines (MVC-NeRF, NeRF VAE and Laser-NV without local geometry) reported were trained with depth supervision, i.e. exactly the same way as Laser-NV was trained. We hope this addresses your concern regarding comparison fairness.
> > > > >
> > > > > To summarize our discussion:
> > > > > - In the only dataset we've used depth supervision for efficiency, all models are trained the same way and tested without depths.
> > > > > - Additionally, Laser-NV performance is very similar when trained without depth as per our supplementary results.
> > > > > - We have included MultiShapenet results that are trained without depth; these are also state-of-the-art and corroborates that our model can learn well without depths.
> > > > > - Finally, we will report the main results on City by training without depth before the end of the discussion period.

---

> > > > > ### Author Response · Authors · 2022-12-09
> > > > > **Results on City without depth supervision**
> > > > >
> > > > > As per your suggestion, we have now run the main experiments on City (Table 1.) without using depth supervision. These are the results:
> > > > > ```
> > > > >                  Rec. PSNR   Rec. SSIM     IWAE |  FID
> > > > > MVC-NeRF:        29.76        0.936           -    81.50
> > > > > NeRF VAE:        23.49        0.658         3.44   76.13
> > > > > Laser (no geom)  27.21        0.866         3.84   24.93
> > > > > Laser-NV:        30.74        0.943         3.87   21.70
> > > > > ```
> > > > >
> > > > > Laser-NV performs considerably better than all the other baselines under all reported metrics, and in general the results are consistent with and very similar to (if not slightly better than) the results of the models trained without depth supervision.The only exception is NeRF VAE whose results are worse than before. Note that we do not yet have results for the version of NeRF VAE  with a conditional prior (cond. NeRF VAE). We aim to include those results too in the final version.
> > > > > Let us know if there are any other questions regarding the use of depth.

---

### Author Response · Authors · 2022-11-16
**Shared Response**

We thank the reviewers for their valuable feedback.

The reviewers agree that the paper is clearly written (TTqh, K538, b7vH) and easy to follow (TTqh, K538), with enough detail to reproduce our methods (b7vH). They also note that LASER-NV improves over the previous state-of-the-art generative model (NeRF-VAE) in terms of reconstructions (less blurry; TTqh, b7vH, KWNb) and novel view synthesis (b7vH). The reviewers also appreciate our experimental design (TTqh, b7vH).

## Novelty
KWNb and K538 find our set representation and the permutation-invariant normalizing flow novel and interesting. This is in contrast to TTqh and b7vH who claim that the novelty of our paper is limited, especially given NeRF-VAE and SRT.

We would like to point out that Laser’s latent set representation, while superficially similar to SRT and NeRF-VAE, is actually more related to Perceiver [1]. To see this, note that the set tokens in SRT/NeRF-VAE both have spatial meaning and are directly derived from the input images. This is generally not true for Perceiver or for our model. Note that, to the best of our knowledge, there have been no previous scene models using a Perceiver-style representation.

Moreover, the main novelty of our paper is a probabilistic treatment of that set, i.e. the modeling of  a conditional prior/posterior distribution which covers multiple modes (unlike any existing scene model). To our knowledge, there is no scene model that
a) uses a probabilistic latent set,
b) is able to sample diverse predictions of previously unseen parts of a scene while being consistent with observations,
c) has state-of-the-art reconstruction performance on challenging datasets (i.e. matching SRT performance on MSN-hard)

From a methodology perspective, Laser-NV includes novel modeling for conditional set-valued, permutation invariant normalizing flows (see Appendix A.1). We have clarified this in the list of contributions (Sec. 1).

## Too Simple Data, No Real Data and SRT Comparison (as per TTqh’s request)
KWNb, b7vH would like to see our method evaluated on a more complex, ideally real-world dataset such as those used in NeRF. This would demonstrate that our model scales to complex data (b7vH) and that it can handle noisy real-world cameras (KWNb).

We now updated the manuscript with the MultiShapeNet (MSN) data used in SRT, and show that we match SRT’s performance on this challenging benchmark. At the same time Laser-NV additionally provides the discussed generative modeling capabilities. Note that MSN is considerably harder than other datasets used in generative scene models, and that Laser-NV is the first of its class to successfully model this data. We believe that Laser, with its competitive results on MSN, constitutes a big step towards generative modeling of real-world data.

We comment on using noisy cameras in the response to KWNb.

## The intuition behind the latent set representation and its interpretation
The decision of using a latent set representation was driven by the observation that:
The cost of increasing the dimensionality of a vector-valued latent variable grows quadratically and that doing so has a performance ceiling (Fig 5b).
As TTqh notices, some models use a spatially-organized data structure that is often sparse.

We were therefore looking for a structure that would allow for high capacity, spatial organization, and that is easy to predict. A voxel grid is usually high-dimensional and very memory-consuming to predict with an encoder. A sparse version thereof is difficult to work during training  (the sparsity pattern would be different for different minibatch items). A set matches these requirements and is well-studied in the literature due to the success of the transformer architecture. Additionally, it is possible for either spatial or semantic structure to emerge in a latent set. Since the attention queries are derived from 3D positions, the set elements could assume the role of voxels in a sparse voxel-like representation with vectors located in densely-occupied regions of the scene.

## Updates to the manuscript
* New experimental results on the challenging MSN-Hard dataset, including qualitative samples from our model (Sec. 4 and Appendix C) and a gif in the supp. material.
* Included results and discussions using cameras with different levels of noise (Appendix D).
* Further discussed the novelty of our latent set representation and model in the contributions (Sec. 1).
* We’ve extended the discussion on the limitations of the model to address the lack of real data results (Sec. 6).
* We’ve included an equation that defines our normalizing flow in Sec. 3.1 to clarify how the latents are modeled and sampled.

### References
1. Jaegle et al. Perceiver: General Perception with Iterative Attention. ICML 2021

---

> ### Comment · Reviewer_TTqh · 2022-12-01
> **Comments regarding the response**
>
> I thank the authors for their considerate response. Below are the comments regarding the responses.
>
> &nbsp;
>
> ### Novelty
>  I agree with the authors that the proposed method is able to sample diverse predictions as well as achieve state-of-the-art performance. The method shows improvements in generative modeling from multiple views.
>
> However, I still think the method is quite similar to SRT and Nerf-VAE.  Also, the claim that the method is more related to Perceiver model is slightly weak considering the manuscript does not mention the Perceiver model.
>
> &nbsp;
> ### Additional Experiments
> I thank the authors for the hard work. The provided additional experiments strengthen the paper.
>
> &nbsp;
> ### Intuition behind the latent set representation and its interpretation
> I agree with the authors that a spatial or semantic structure can emerge from latent set representation. The arguments would have been alot stronger if a theory behind the method or experimental results such as interpolating the latent space were provided.
> Also, I do not think a sparse voxel version is difficult to work with considering using many spatially sparse, differentiable libraries from recent works [1, 2].
>
>
> &nbsp;
>
> [1] Choy et al. 4D Spatio-Temporal ConvNets: Minkowski Convolutional Neural Networks. CVPR, 2019
>
> [2] Fey, Mathhias, pytorch-scatter. https://github.com/rusty1s/pytorch_scatter

---

> > ### Author Response · Authors · 2022-12-05
> > **Novelty & Set Intuition**
> >
> > ## Similarity to Perceiver
> > We agree with the reviewer that our method is superficially similar to SRT and NeRF-VAE. However, we think the representation Laser-NV uses is more related to Perceiver.
> >
> > We agree with the reviewer that we should (and will) cite the Perceiver work. By referring to Perceiver we simply intended to illustrate the key mechanism that both Laser-NV and Perceiver have that clearly differs from SRT's mechanism.
> > SRT and Laser-NV are substantially different in two ways. First, in the shape and meaning of the sets: SRT has a 1-to-1 correspondence between image features and latent features (their latents are spatially localized).
> > In Laser-NV, just like in Perceiver, the set tokens do not have a spatial meaning. Also the size of the set is independent of the input size (see Fig. 15 in Appendix D for results with different set sizes used at test time). This gives Laser the flexibility to choose how much latent capacity to allocate for a scene, something SRT cannot do.
> >
> > Second, the probabilistic treatment of our latent representation means the two methods have fundamentally different capabilities (multimodal sampling vs single prediction) and architectural design (conditional normalizing flow vs Transformer encoder).
> > **For the reasons above, Laser-NV is not a simple combination of NeRF VAE and SRT, it has design elements and capabilities that neither of the methods have.**
> >
> > We will update the paper with a citation of Perceiver and an explanation.
> >
> > ## Intuition behind the latent set
> > We do not claim that our set representation has a spatial or semantic structure; that is why these statements do not appear in the paper. We simply provide an intuition of why we wanted to try a set representation, which is what the reviewer asked for. This is independent of the fact that our set representation achieves state-of-the-art performance and scales well, which should be enough to justify its use.
> >
> > Thank you for providing pointers to the libraries with support for sparse computation. Unfortunately, the hardware we use does not support these operations. Consequently, there is a benefit in developing approaches that are not tied to specific hardware (in this case GPUs). The design choice of using a set as opposed to any other form of representation should not be held against us.
> >
> > ## Additional Experiments
> > If the reviewer is satisfied with the additional experiments, could we please ask for a score upgrade? Thank you.

---

### Decision · Program_Chairs · 2023-01-20

**Decision:**

Reject

**Justification For Why Not Higher Score:**

Lack of real world results.

**Justification For Why Not Lower Score:**

N/A

**Metareview: Summary, Strengths And Weaknesses:**

The paper proposes a  set-valued latent representation using normalizing flows  in a multi view transformer prediction model. By using a latent set representation instead of a single vector as Nerf-Vaes, the results are way better, and can generate multi-modal predictions. The reviewers raised concerns regarding comparisons with SRT which also uses a set-valued latent representation, where the latent are in 1-to-1 correspondence with patches in the multi-view images, and the authors added the requested experiments. The reviewers further asked about experiments on real world data such as in the StreetView benchmark, but the authors said that dynamic entities are part of the dataset and they cannot use it for this purpose. Overall, though reviewers welcomed the additional experiments and explanations, would still like to see application of the model in this real world dataset, even if the results are not the state of the art, to illuminate on the generalization ability of the proposed set representation. For this reason, the authors are encouraged to add the requested experiments and submit to a future venue.

**Summary Of Ac-Reviewer Meeting:**

We concluded the model is novel and that real world experiments are highly desirable.